# High-altitude Hypoxia Influences the Activities of the Drug-Metabolizing Enzyme CYP3A1 and the Pharmacokinetics of Four Cardiovascular System Drugs

**DOI:** 10.3390/ph15101303

**Published:** 2022-10-21

**Authors:** Junbo Zhu, Yabin Duan, Delong Duo, Jianxin Yang, Xue Bai, Guiqin Liu, Qian Wang, Xuejun Wang, Ning Qu, Yang Zhou, Xiangyang Li

**Affiliations:** 1Research Center for High Altitude Medicine, Qinghai University Medical College, Xining 810000, China; 2State Key Laboratory of Plateau Ecology and Agriculture, Qinghai University, Xining 810000, China; 3Department of Clinical Pharmacy, Qinghai University Affiliated Hospital, Xining 810000, China; 4Department of Anesthesiology, Red Cross Hospital of Qinghai, Xining 810000, China; 5Department of Anesthesiology, Qinghai Hospital of Traditional Chinese Medicine, Xining 810000, China; 6State Key Laboratory of Functions and Applications of Medicinal Plants, Guizhou Provincial Key Laboratory of Pharmaceutics, Guizhou Medical University, Guiyang 550004, China

**Keywords:** cardiovascular system drugs, high-altitude hypoxia, pharmacokinetics, CYP3A1

## Abstract

(1) Background: High-altitude hypoxia has been shown to affect the pharmacokinetic properties of drugs. Although there is a high incidence of cardiovascular disease among individuals living in high-altitude areas, studies on the effect of high-altitude hypoxia on the pharmacokinetic properties of cardiovascular drugs are limited. (2) Methods: The aim of this study was to evaluate the pharmacokinetics of nifedipine, bosentan, simvastatin, sildenafil, and their respective main metabolites, dehydronifedipine, hydroxybosentan, simvastatin hydroxy acid, and N-desmethyl sildenafil, in rats exposed to high-altitude hypoxia. Additionally, the protein and mRNA expression of cytochrome P450 3A1 (CYP3A1), a drug-metabolizing enzyme, were examined. (3) Results: There were significant changes in the pharmacokinetic properties of the drugs in rats exposed to high-altitude hypoxia, as evidenced by an increase in the area under the curve (AUC) and the half-life (t_1/2z_) and a decrease in total plasma clearance (CL_z_/F). However, most of these changes were reversed when the rats returned to a normoxic environment. Additionally, there was a significant decrease in CYP3A1 expression in rats exposed to high-altitude hypoxia at both the protein and mRNA levels. (4) Conclusions: High-altitude hypoxia suppressed the metabolism of the drugs, indicating that the pharmacokinetics of the drugs should be re-examined, and the optimal dose should be reassessed in patients living in high-altitude areas.

## 1. Introduction

High-altitude refers to the area where the altitude exceeds 2500 meters above sea level. High-altitude environments are characterized by high solar radiation, low ambient oxygen tension, extreme diurnal temperature range, arid climate, poor soil quality, and hypoxia [1]. High-altitude hypoxia affects the quality of life of 140 million people living in these areas and individuals who travel to high altitudes for tourism and sports. The digestive, cardiovascular, cerebrovascular system, and nervous systems and exogenous substance metabolism are considerably influenced by high-altitude hypoxia, which can cause high-altitude sickness, including acute mountain sickness, chronic polycythemia, cardiovascular disease, pulmonary edema, and cerebral edema [2]. The cardiovascular system is highly oxygen- and energy-consuming and is sensitive to hypoxia. In particular, the cardiovascular system is affected by oxidative stress, sympathetic nerve excitation, and other mechanisms in high-altitude hypoxic environments, resulting in several physiological and pathological changes [3,4,5,6]. Studies have shown that altitude and the prevalence of hypertension are positively correlated. In areas 3000 m above sea level, the prevalence of hypertension increases by 2% at every 100 meters altitude [7]. Additionally, studies have reported a significantly higher prevalence of hypertension in individuals from plateau areas than in those from plain areas. The risk of hypertension-induced renal damage and coronary heart disease is higher among individuals living in high-altitude areas [8,9,10,11]. Moreover, there is a significantly higher prevalence of cardiovascular diseases among individuals living on plateaus, with a high incidence of arrhythmia, heart failure, pulmonary hypertension, coronary heart disease, hypertension, and other cardiovascular diseases [12,13].

Hypoxia causes a series of physiological changes in body fluids, blood rheology, blood biochemistry, and organ functions, which may affect the absorption, distribution, metabolism, and excretion of drugs [14,15,16]. Studies have shown that the in vivo metabolism of propranolol [17], aminophylline [18], acetazolamide [19], and sulfamethoxazole [20] is significantly affected by hypoxia. Cytochrome P450 (CYP450) is the most important phase I metabolic enzyme, and 57 human CYP450 genes have been identified, which are classified into 18 families and 42 subfamilies according to sequence similarity [21,22,23]. CYP1A2, CYP2C9, CYP2C19, CYP2D6, CYP2E1, and CYP3A4 metabolize more than 90% of drugs, of which CYP3A4 is the most important drug-metabolizing enzyme in the human body, metabolizing more than 50% of prescription drugs [24,25,26]. Simvastatin, sildenafil, bosentan, and nifedipine are metabolized by CYP3A4 in humans and by CYP3A1 in rats [27,28,29,30,31]. Studies have demonstrated that hypoxia significantly alters the activity and expression of the drug-metabolizing enzyme CYP450 [32,33,34,35,36,37,38]. Therefore, it is inferred that CYP3A4 activity may change under hypoxia and then affects the metabolism of these four cardiovascular system drugs.

Simvastatin inhibits endogenous cholesterol synthesis and regulates blood lipid levels [39]. Sildenafil inhibits 5-phosphodiesterase, enhances the nitric oxide/cyclic guanosine monophosphate signaling pathway, increases cardiac blood transfusion, and maintains myocardial function [40,41]. Bosentan, an endothelin receptor antagonist, is the first-line drug for treating pulmonary hypertension [42]. Nifedipine is the first dihydropyridine calcium channel blocker that regulates blood pressure by relaxing blood vessels and reducing afterload. Nifedipine is widely used in the clinical treatment of hypertension and angina pectoris [27]. However, the pharmacokinetics of simvastatin, sildenafil, bosentan, and nifedipine in high-altitude hypoxic environments are limited. Therefore, the aim of this study was to evaluate the pharmacokinetics of these drugs under high-altitude hypoxia in rats and examine the protein and mRNA expression levels of CYP3A1 to provide information on rational drug use at high altitudes.

## 2. Results

### 2.1. UHPLC-MS Method Validation

Under the current analytical conditions, the concentrations of the four drugs and their metabolites peaked within the analysis duration. The peaks of nifedipine, bosentan, simvastatin, sildenafil, dehydronifedipine, hydroxybosentan, simvastatin hydroxy acid, and N-desmethyl sildenafil were well differentiated, and no endogenous interference was observed in rat plasma.

Excellent linearity was observed for simvastatin, simvastatin hydroxy acid, sildenafil, N-desmethyl sildenafil, nifedipine, dehydronifedipine, bosentan, and hydroxybosentan within the ranges of 3.91–250, 1.95–125, 0.16–83.32, 0.65–41.66, 5–200, 0.1–5, 5–200, and 0.156–10 ng/mL, respectively, with the coefficient of correlation exceeding 0.9997, 0.9993, 0.9987, 0.9988, 0.9995, 0.9900, 0.9964, and 0.9953, respectively. Both intraday and interday precisions for all drugs and their metabolites were < 15%, and accuracy ranged from 85 to 115%. The RSD of stability and matrix effect for all drugs and their metabolites were < 15%. Recovery rates of 102.54, 104.43, and 102.70% were achieved for low, medium, and high concentrations of simvastatin control samples, respectively; 101.37%, 102.22%, and 103.05% for low, medium, and high concentrations of simvastatin hydroxy acid, respectively; 101.34%, 104.12%, and 102.99% for low, medium, and high concentrations of sildenafil, respectively; 102.87%, 102.55%, and 105.12% for low, medium, and high concentrations of N-desmethyl sildenafil, respectively; 99.84%, 103.08%, and 105.39% for low, medium, and high concentrations of nifedipine, respectively; 101.77%, 105.26%, and 104.61% for low, medium, and high concentrations of dehydronifedipine, respectively; 103.63%, 101.66%, and 101.04% for low, medium, and high concentrations of bosentan, respectively; and 101.56%, 103.98%, and 102.30% for low, medium, and high concentrations of hydroxybosentan, respectively.

### 2.2. Physiologic and Biochemical Parameters

Changes in hematological and serum parameters of rats exposed to high-altitude hypoxia are shown in Figure 1 and Figure 2. The hemoglobin levels of rats in acute hypoxia, chronic hypoxia, and chronic hypoxia to normoxia groups increased by 11.59, 38.62, and 24.86%, respectively, whereas their hematocrit levels decreased by 34.56, 23.36, and 18.89%, respectively, compared with those of rats in the normoxia group. Additionally, there was a 28.24, 29.89, and 36.37% decrease in the mean corpuscular volume of rats in the acute hypoxia, chronic hypoxia, and chronic hypoxia to normoxia groups, respectively, compared with those of rats in the normoxia group. Similarly, there was a 29.56, 22.22, and 15.11% decrease in the mean platelet volume of rats in the acute hypoxia, chronic hypoxia, and chronic hypoxia to normoxia groups, respectively, compared with those of rats in the normoxia group. In contrast, the red blood cell count of rats in the acute hypoxia and chronic hypoxia groups increased by 9.32 and 27.27% compared with that of the normoxia group, whereas that of rats in the chronic hypoxia to normoxia group decreased by 8.62%. Similarly, there was a 51.81% increase in the white blood cell count of rats in the acute hypoxia group and a 19.60% decrease in the platelet count of rats in the chronic hypoxia group compared with that of the rats in the normoxia group. There were no significant changes in the levels of other hematological parameters.

The albumin level of rats in the acute hypoxia, chronic hypoxia, and chronic hypoxia to normoxia groups increased by 39.41, 30.34, and 24.77%, respectively, while the glucose levels increased by 58.75, 96.65, and 66.50%, respectively, compared with those of the rats in the normoxia group. In contrast, the alanine aminotransferase levels of rats in the acute hypoxia, chronic hypoxia, and chronic hypoxia to normoxia groups decreased by 58.25, 51.32, and 48.74%, respectively, while aspartate aminotransferase levels decreased by 61.13, 58.00, and 47.95%, respectively, compared with those of the rats in the normoxia group. Similarly, the total protein levels of rats in the acute hypoxia, chronic hypoxia, and chronic hypoxia to normoxia groups decreased by 13.74, 11.41, and 13.98%, respectively, while the globulin levels decreased by 48.27, 38.43, and 39.15%, respectively, compared with those of rats in the normoxia group. Additionally, there was a 25.61 and 33.62% decrease in the serum urea levels of rats in the acute hypoxia and chronic hypoxia groups, respectively, and an 8.97% increase in the serum triglyceride levels of rats in the chronic hypoxia to normoxia group compared with those of the rats in the normoxia group. The raw data of physiologic and biochemical parameters is shown in Appendix A.

### 2.3. Hematoxylin-Eosin Staining of Heart, Kidney, Liver, and Lung Tissues in Rats

A high-altitude hypoxic environment has been shown to affect the heart, kidney, liver, and lung tissue of rats. The HE staining results showed no noticeable inflammatory changes in any of the tissues of the rats in the normoxia group; specifically, the myocardial cells were clearly demarcated, the horizontal stripes were conspicuous, and the light and dark were alternated. The renal tubular epithelial cells were round and plump, brush borders were neat and regular, and the medulla showed no apparent abnormalities. The liver and lung tissue membranes were intact, liver plates were regularly arranged, and alveolar walls were not significantly thickened. In the acute and chronic hypoxia groups, minor changes were observed in various tissues. The interstitial capillaries in the heart and kidney tissues were slightly expanded, tiny local vacuoles were observed in the liver cells, and the alveolar walls were slightly thickened. The alveolar space was narrowed, and there was no apparent inflammatory cell infiltration in any of the tissues. After the rats in the chronic hypoxia group were returned to the normoxic environment, the effects of hypoxia on the heart, kidneys, and liver tissues were reduced to a certain extent. A few capillaries in the heart and kidney tissues were dilated, and the fat vacuoles in the liver cytoplasm became small. However, the alveolar wall was still slightly thickened, comparable to that of the rats in the hypoxia group; however, no apparent parenchymal damage was found in any tissue (Figure 3).

### 2.4. Pharmacokinetics

The mean pharmacokinetic parameters of simvastatin, simvastatin hydroxy acid, sildenafil, N-desmethyl sildenafil, nifedipine, dehydronifedipine, bosentan, and hydroxybosentan are shown in Table 1, Table 2, Table 3, Table 4, Table 5, Table 6, Table 7 and Table 8, respectively. The mean plasma concentration-time profiles for simvastatin, simvastatin hydroxy acid, sildenafil, N-desmethyl sildenafil, nifedipine, dehydronifedipine, bosentan, and hydroxybosentan are shown in Figure 4, Figure 5, Figure 6, Figure 7, Figure 8, Figure 9, Figure 10 and Figure 11, respectively. There was a 31.13 and 22.03% increase (*p* < 0.05) in the t_1/2z_ of simvastatin in the plasma of rats in the chronic hypoxia group compared with those of rats in the normoxia and acute hypoxia groups, respectively. Similarly, the MRT_0-t_ of simvastatin in the plasma of rats in the acute and chronic hypoxia to normoxia groups were 13.18 and 11.95% higher (*p* < 0.05) than that of the rats in the normoxia group, respectively. Although the data were not statistically significant, there was a 39.24, 56.65, and 53.02% increase (*p* > 0.05) in AUC_0-t_ in the acute hypoxia, chronic hypoxia, and chronic hypoxia to normoxia groups, respectively, compared to that in the normoxia group, whereas there was a 27.54, 39.58, and 26.4% decrease (*p* < 0.05) in CL_z_/F values, respectively. Additionally, the C_max_ value of the chronic hypoxia group was 44.90% higher (*p* < 0.05) than that of the normoxia group. However, the groups had no significant differences (*p* > 0.05) in the V_z_/F and Tmax values. Although the data were not statistically significant, the t_1/2z_ of simvastatin hydroxy acid in the plasma of the rats was 6.77, 16.28, and 11.50% higher in the acute hypoxia, chronic hypoxia, and chronic hypoxia to normoxia groups, respectively, compared with that in the normoxia group. There was a 47.62 and 37.87% increase (*p* < 0.05) in the MRT_0-t_ of simvastatin hydroxy acid in the plasma of rats in the chronic hypoxia and chronic hypoxia to normoxia groups, respectively, compared with that in the normoxia group, and a 38.17 and 29.05% increase (*p* < 0.05), respectively, compared with that in the acute hypoxia group. There was a 42.48% decrease (*p* < 0.05) in the V_z_/F value of the chronic hypoxia group compared with that in the normoxia group. Compared with the normoxia group, CL_z_/F values were 50.33% and 30.42% lower (*p* < 0.05) in the chronic hypoxia and chronic hypoxia to normoxia groups, respectively. Similarly, there was a 120.21% and 57.85% increase in AUC_0-t_ in the chronic hypoxia and chronic hypoxia to normoxia groups (*p* < 0.05), respectively, compared with that in the normoxia group. Additionally, C_max_ values were 79.05 and 53.05% (*p* < 0.05) higher in the chronic hypoxia and chronic hypoxia to normoxia groups, respectively, compared with that in the normoxia group. However, there was no significant difference in T_max_ among the groups.

Although the data were not statistically different, the t_1/2z_ values for sildenafil were 6.79, 14.29, and 11.53% higher in the acute hypoxia, chronic hypoxia, and chronic hypoxia to normoxia groups, respectively, compared with that in the normoxia group. The MRT_0-t_ of the drug was 15.55% lower in the chronic hypoxia to normoxia group (*p* < 0.05) than that in the normoxia group, and 17.25% (*p* < 0.05) and 22.07% (*p* < 0.05) lower in the chronic hypoxia to normoxia group than that in the acute and chronic hypoxia groups. The V_z_/F of the drug was 40.56, 38.24, and 34.07% lower in the acute hypoxia (*p* < 0.05), chronic hypoxia (*p* < 0.05), and chronic hypoxia to normoxia (*p* < 0.05) groups, respectively, compared with that in the normoxia group. Similarly, there was a 45.29, 45.98, and 41.11% decrease (*p* < 0.05) in the CL_z_/F of the drug in the acute hypoxia, chronic hypoxia, and chronic hypoxia to normoxia groups, respectively, compared with that in the normoxic group. In contrast, there was a 69.69, 96.56, and 74.59% increase (*p* < 0.05) in the AUC_0-t_ in the acute hypoxia, chronic hypoxia, and chronic hypoxia to normoxia groups, respectively, compared with that in the normoxia group. Similarly, there was an 80.50, 109.98, and 110.07% increase (*p* < 0.05) in the C_max_ of the drug in the acute hypoxia, chronic hypoxia, and chronic hypoxia to normoxia groups, respectively, compared with that in the normoxia group. However, there was no significant difference in the T_max_ of the drug among the groups. There was a 2.70% increase (*p* < 0.05) in the t_1/2z_ of N-desmethyl sildenafil in the chronic hypoxia group (*p* < 0.05) compared with the normoxia group; however, there was no significant difference in the t_1/2z_ values between the acute hypoxia (3.09% increase) and chronic hypoxia to normoxia (7.11% increase) groups and the normoxia group. Additionally, there was a 10.24% decrease and a 16.84% increase in the MRT_0-t_ of the metabolite in the acute hypoxia (*p* < 0.05) and chronic hypoxia groups (*p* < 0.05), respectively, compared with that in the normoxia group. Moreover, there was a 37.73% decrease in the V_z_/F of the metabolite in the chronic hypoxia group (*p* < 0.05) compared with that in the normoxia group. The CL_z_/F of the metabolite were 38.63 and 28.44% lower in the chronic hypoxia and chronic hypoxia to normoxia groups (*p* < 0.05), respectively, than in the normoxia group. In contrast, the AUC_0-t_ increased by 81.44 and 38.57% in the chronic hypoxia and chronic hypoxia to normoxia groups (*p* < 0.05), respectively, compared with that in the normoxia group. Similarly, the C_max_ of the metabolite increased (*p* < 0.05) by 83.07 and 62.28% in the chronic hypoxia and chronic hypoxia to normoxia groups, respectively, compared with that the normoxia group. However, there was no significant difference in the T_max_ of the metabolite among the groups.

The t_1/2z_ of nifedipine in the plasma increased by 19.45, 23.95, and 23.45% in the acute hypoxia (*p* < 0.05), chronic hypoxia (*p* < 0.05), and chronic hypoxia to normoxia groups (*p* < 0.05), respectively, compared with that in the normoxia group. Similarly, there was a 17.58 and 20.51% increase in the MRT_0-t_ of the drug in the chronic hypoxia and chronic hypoxia to normoxia groups (*p* < 0.05), respectively, compared with that in the normoxia group. In contrast, the V_z_/F of the drug increased by 23.44 and 24.95% in the chronic hypoxia and chronic hypoxia to normoxia groups (*p* < 0.05), respectively, compared with that in the normoxia group. Compared with the normoxia group, there was an 18.51, 38.80, and 39.76% decrease in the CL_z_/F of nifedipine in the acute hypoxia, chronic hypoxia, and chronic hypoxia to normoxia groups (*p* < 0.05), respectively. The AUC_0-t_ increased (*p* < 0.05) by 19.47, 56.08, and 58.56% in the acute hypoxia, chronic hypoxia, and chronic hypoxia to normoxia group compared with that in the normoxia group. Compared with the normoxia group, there was a 32.51% increase (*p* < 0.05) in the C_max_ of nifedipine in the chronic hypoxia group. However, there was no significant difference in T_max_ among the groups. Although the data were not statistically different, the t_1/2z_ of dehydronifedipine in the plasma increased by 19.89, 31.81, and 21.04% in the acute hypoxia, chronic hypoxia, and chronic hypoxia to normoxia groups, respectively, compared with that in the normoxia group. Similarly, the MRT_0-t_ of dehydronifedipine increased by 8.60, 18.67, and 16.97% in the acute hypoxia (*p* < 0.05), chronic hypoxia (*p* < 0.05), and chronic hypoxia to normoxia groups (*p* < 0.05), respectively, compared with that in the normoxia group. In contrast, the V_z_/F of dehydronifedipine decreased by 48.85 and 41.00% in the chronic hypoxia and chronic hypoxia to normoxia groups (*p* < 0.05), respectively, compared with that in the control group. Similarly, there was a 32.77, 61.33, and 51.03% decrease in the CL_z_/F of the metabolite in the acute hypoxia, chronic hypoxia, and chronic hypoxia to normoxia groups (*p* < 0.05), respectively, compared with that in the normoxia group. In contrast, the AUC_0-t_ increased (*p* < 0.05) by 44.65, 146.76, and 99.72% in the acute hypoxia, chronic hypoxia, and chronic hypoxia to normoxia groups, respectively, compared with that in the normoxia group. Similarly, there was a 72.30 and 38.79% increase (*p* < 0.05) in the C_max_ of dehydronifedipine in the chronic hypoxia and chronic hypoxia to normoxia groups, respectively, compared with that in the normoxia group. Additionally, the T_max_ of dehydronifedipine was 27.59% higher (*p* < 0.05) in the acute hypoxia group than that in the normoxia group.

There was no significant difference in the t_1/2z_ of bosentan among the four groups. However, there was a 10.57 and 14.28% increase in the MRT_0-t_ of bosentan in the chronic hypoxia (*p* < 0.05) and chronic hypoxia to normoxia groups (*p* < 0.05), respectively, compared with that in the normoxia group. The V_z_/F of bosentan in the plasma of the rats decreased (*p* < 0.05) by 44.39% in the chronic hypoxia group compared with that in the normoxia group. Similarly, there was a 54.12 and 40.18% decrease (*p* < 0.05) in the CL_z_/F of bosentan in the chronic hypoxia and chronic hypoxia to normoxia groups, respectively, compared with that in the normoxia group. The AUC_0-t_ increased by 104.22 and 59.31% in the chronic hypoxia and chronic hypoxia to normoxia groups (*p* < 0.05), respectively, compared with that in the normoxia group. Similarly, there was a 64.02% increase (*p* < 0.05) in the C_max_ of bosentan in the plasma samples of rats in the chronic hypoxia group compared with that in the normoxia group. However, there was no significant difference in the T_max_ of bosentan among the groups. Furthermore, there were no significant differences in the t_1/2z_ and MRT_0-t_ of hydroxybosentan in the plasma samples among the four groups. However, there was a 62.70% decrease (*p* < 0.05) in the V_z_/F of hydroxybosentan in the chronic hypoxia group compared with that in the normoxia group. Similarly, there was a 35.21, 66.54, and 43.58% decrease in the CL_z_/F of hydroxybosentan in the acute hypoxia, chronic hypoxia, and chronic hypoxia to normoxia groups (*p* < 0.05), respectively, compared with that in the normoxia group. The AUC_0-t_ increased by 54.38, 184.66, and 93.27% in the acute hypoxia (*p* < 0.05), chronic hypoxia (*p* < 0.05), and chronic hypoxia to normoxia groups (*p* < 0.05), respectively, compared with that in the normoxia group. Compared to the normoxia group, there was a 121.87 and 90.17% increase in the C_max_ of hydoxybossentan in the chronic hypoxia and chronic hypoxia to normoxia groups (all *p* < 0.05), respectively. However, there was no significant difference in the T_max_ of hydoxybossentan among the four groups.

### 2.5. Protein Expression of CYP3A1

There was a significant decrease in the protein expression of CYP3A1 in the liver of rats exposed to hypoxic conditions. There was a 23.95, 36.46, and 29.59% decrease in the protein expression of CYP3A1 in the acute hypoxia, chronic hypoxia, and chronic hypoxia to normoxia groups (*p* < 0.05), respectively, compared with the normoxia group (Figure 12). Additionally, CYP3A1 protein expression of the chronic hypoxia group was 16.45% lower (*p* < 0.05) than that of the acute group; however, no significant differences were observed between the remaining groups. The original pictures of Western blotting is shown in Appendix A. 

### 2.6. mRNA Expression of CYP3A1

There was a significant decrease in the mRNA expression of CYP3A1 in rats exposed to high-altitude hypoxia. Compared to the normoxia group, there was a 52.23, 64.74, and 57.59% decrease (*p* < 0.05) in the CYP3A1 mRNA expression levels in the acute hypoxia, chronic hypoxia, and chronic hypoxia to normoxia groups(Figure 13). However, no significant differences were observed between the remaining groups. The qPCR raw data of CYP3A1 is shown in Appendix A.

## 3. Discussion

The body undergoes several physiological and pathological changes under high-altitude hypoxic conditions, among which hematological and serum biochemical parameters are important indicators of the health status of individuals. As one of the body’s defense systems against foreign bodies, white blood cell count can reflect the immune status of the body. The results of the present study indicated an increase in the white blood cell count of rats in both the acute and chronic hypoxia groups, indicating that hypoxic conditions may induce inflammatory responses, which was inconsistent with a previous finding [43] that reported a decrease in the white blood cell count of rats in a hypoxic environment. The differences in results could be attributed to differences in study methodology, hypoxia modeling method, modeling time of rats, and the noncontinuous hypoxia state and hypoxia time caused by the need for animal treatment during the simulated hypoxia modeling period. Hemoglobin count and oxygen transport in the body increase under hypoxic conditions to meet oxygen demand. Consistent with previous reports [44], the results of the present study showed an increase in the hemoglobin count of rats exposed to hypoxic environments, which could be attributed to the increase in oxygen demand by the body in response to hypoxia-induced oxidative stress. High-altitude hypoxic environments have been shown to increase uric acid production, which is directly related to the number of red blood cells [45]. The results of the present study showed an increase in the red blood cell count of rats under high-altitude hypoxic conditions; therefore, it was speculated that the increased red blood cell count could cause hyperuricemia. Moreover, a high red blood cell count could cause an increase in blood viscosity [46], which can suppress blood flow and inhibit circulation, resulting in myocardial damage [47,48]. Platelets are released by mature megakaryocytes in the bone marrow, and changes in platelets are mainly due to an imbalance in their production and destruction. Previous studies have reported a decrease in platelet count after long-term exposure to high-altitude environments, which is consistent with the results of the present study. However, platelet count gradually recovers when the individual is returned to normoxic conditions (plain areas) [49], and this change is related to the body’s pulmonary arterial pressure, hypoxic stress state, estrogen levels, and other factors [50,51].

Biochemical indicators can reflect changes in blood lipids, liver function, and renal function to a certain extent. The results of the present study showed that the serum biochemical parameters of the rats were significantly affected compared to those in the normoxic group, except for serum urea and high-density lipoprotein contents of rats in the chronic hypoxia to normoxia group. The findings of the present study showed that hypoxia had significant effects on blood lipid levels and liver and kidney functions in rats. Changes in liver function can impact the production and efficacy of drug-metabolizing enzymes, thereby affecting the metabolism of drugs in the body [14]. The decrease in the urea concentration in rats exposed to hypoxic conditions indicates that the body’s ability to excrete drugs is affected under high-altitude hypoxic conditions, resulting in an increased risk of adverse reactions. Additionally, the results of the present study showed that rats exposed to hypoxic conditions had higher glucose levels and lower triglyceride and cholesterol levels than those exposed to normoxic conditions. Similarly, Shin [52] reported an increase in the blood sugar level of rats exposed to hypoxic conditions intermittently, which was consistent with the results of the present study. However, most of the biochemical parameters returned to normal after the rats were returned to normoxic conditions (plain areas), indicating that the effect of high-altitude hypoxia on biochemical indicators was reversible. This validates the application of oxygen therapy for patients in high-altitude areas.

Under high-altitude hypoxic conditions, there is an increase in the generation of oxygen free radicals by the body, inducing lipid peroxidation, which affects the structure and functions of various organs [53]. Similarly, the histochemical analysis in this study showed that hypoxia could affect local capillaries in rat myocardium. Changes in the electrocardiograms of rats under hypoxic conditions were observed in a previous study, with a decrease in the rate of myocardial depolarization and repolarization, further inducing myocardial damage in rats [54]. Additionally, previous studies have shown that high-altitude hypoxia affects renal function [53,55]. Specifically, significantly larger glomeruli were observed in children living in high-altitude environments compared with those of children living in plain areas, which may be related to hyperuricemia and activation of renin–angiotensin [53]. Additionally, there is an increase in blood renin and angiotensin levels of patients with kidney injury. Therefore, angiotensin-converting enzyme inhibitors or AT1 receptor antagonists could be used as clinical interventions to alleviate the effect of high-altitude hypoxia on the kidneys [56]. The liver is the most important metabolic organ in the body and is highly sensitive to hypoxia. In the early stages of hypoxia, the body provides energy and fights hypoxia by enhancing glycolysis, increasing the surface area of cell line membranes, and increasing the number of key enzymes in the tricarboxylic acid cycle. However, once the degree of hypoxia exceeds the body’s ability to compensate, these adaptive changes become unbalanced. First, the tissue morphology of the liver changes, the permeability of the liver cell membrane increases due to the infiltration of inflammatory cells, and the microcirculation is blocked. These changes are accompanied by a decrease in the energy level of liver cells, resulting in low immune and abnormal liver functions. High-altitude hypoxia and cold environments affect the structure and function of lung tissue. Oxygen free radicals accumulated in the lungs can damage capillary endothelial cells and increase vascular permeability [57], resulting in dyspnea and increased respiratory rate [58,59]. The results of the present study showed that there was only a slight thickening of the alveolar walls of rats in the hypoxia group, with no apparent pulmonary edema, indicating that the high-altitude environment in this study did not cause severe damage to the lung tissue. Additionally, changes in the heart, liver, lung, and kidney tissues of rats exposed to hypoxic conditions were alleviated after they were returned to normal conditions (plain areas); however, there were still some differences compared to those from the normoxia group. These results indicate that hypoxia-induced changes in organ structure and functions may not be completely reversible after returning to normal environments (plain areas), which should be considered during drug administration for hypoxia.

Furthermore, chronic and acute hypoxic conditions significantly affected the pharmacokinetic parameters of nifedipine, bosentan, simvastatin, and sildenafil. Compared to those in the normoxia groups, there was an increase in the AUC and a decrease in the CL_z_/F for the drugs in rats in the hypoxic group. Similarly, there was an increase in the t_1/2z_ of the drugs in the plasma of rats in the acute and chronic hypoxia groups compared with those in the normoxia group, indicating that hypoxia suppressed the excretion of nifedipine, bosentan, simvastatin, and sildenafil. Dehydronifedipine, hydroxybosentan, simvastatin hydroxy acid, and N-desmethyl sildenafil are the major metabolites of nifedipine, bosentan, simvastatin, and sildenafil, respectively, at high altitudes, and their metabolic kinetics were also affected by hypoxic environments. Similar to the results of the drugs, there was an increase in the AUC and CL_z_/F of these metabolites under hypoxic conditions. Additionally, there was an increase in the t_1/2z_ of the metabolites in the plasma of rats in the three hypoxia groups, confirming that the metabolism of the four drugs was slowed under hypoxia. 

As the most important enzyme system in the liver, cytochrome P450 metabolizes most exogenous substances [60,61]. Therefore, pharmacokinetic studies under high-altitude hypoxia are mainly focused on the effect of hypoxia on the activity of the CYP450 enzyme [14]. Nifedipine, bosentan, simvastatin, and sildenafil are metabolized mainly by CYP3A4 in the human liver and CYP3A1 in rat liver [38,39,40,41,42]. The present study showed a significant decrease in the expression of CYP3A1 in rats after exposure to acute and chronic hypoxia at both the protein and mRNA levels. The changes in the pharmacokinetics of nifedipine, bosentan, simvastatin, and sildenafil in rats after exposure to high-altitude hypoxia may be attributed to a significant decrease in CYP3A1 expression. Similarly, the results of our previous studies showed a decrease in the activity and protein and mRNA expression of CYP1A2 in rats exposed to a hypoxic environment in a 4300 m plateau; however, there were no significant changes in the expression of CYP2D1, CYP2E1, CYP3A1, and CYP2C11. Additionally, chronic hypoxia significantly affected the activity and protein and mRNA expression of CYP1A2, CYP2D1, CYP2E1, and CYP3A1 in rats [21,22]. Fradette [62,63] simulated a high-altitude field environment in a hypobaric oxygen chamber and observed a decrease in the protein expression of CYP1A1, CYP1A2, CYP2B4, CYP2C5, and CYP2C16, but an increase in the protein expression of CYP3A6 and CYP3A11 in rabbits. The results of studies on the effect of high-altitude hypoxia on protein and mRNA expression of CYP2C11 have been inconsistent [21,64,65]. Therefore, it is crucial to examine the response of each enzyme of the CYP450 family to high-altitude hypoxia.

Recently, considerable progress has been made on the mechanism of hypoxia on drug metabolism. As transcription factors, nuclear receptors, including pregnane X receptor (PXR), constitutive androstane receptor (CAR), and peroxisome proliferator activation receptor (PPAR), play an important role in regulating drug-metabolizing enzymes [65]. PXR and CAR have been shown to regulate the expression of CYP2C9, CYP2C18, and CYP2C19 [65]. Moreover, there is a considerable decrease in PXR and CAR expression under hypoxic conditions [66]. Hypoxic conditions can inhibit the expression of CYP1A2, CYP2C9, CYP2E1, CYP3A4, and UGT1A1 through the PXR and CAR regulatory pathways based on dual-luciferase assays [67]. miRNAs are a class of small noncoding RNAs with a length of approximately 19–25 nucleotides. Hypoxic conditions can cause considerable changes in miRNA expression, affecting the expression and activities of drug-metabolizing enzymes and nuclear receptors. Additionally, changes in genes related to drug metabolism indirectly regulate the expression of drug-metabolizing enzymes [65]. In a previous study, we observed upregulation in the expression of 27 miRNAs and downregulation in the expression of 56 miRNAs in Caco-2 cells exposed to hypoxia. Further analysis confirmed that hypoxia led to a significant decrease in the expression of miRNA-873-5p; moreover, miRNA-873-5p possesses a similar effect as PXR on drug-metabolizing enzymes [68].

The gut microbiota is a complex ecosystem of microorganisms that inhabits the gastrointestinal tract. Recently, studies have focused on the role of the gut microbiota in drug metabolism [69]. Microbial enzymes and intestinal flora metabolites have been shown to affect drug metabolism directly, and the intestinal flora can indirectly mediate drug metabolism by changing the expression of host phase I and II drug-metabolizing enzymes [70]. Zhang [71] found that an imbalance of intestinal flora in high-altitude hypoxic environments can reduce the metabolic rate of nifedipine. Additionally, gene sequencing and bioinformatics analyses have shown that the structure and diversity of the intestinal flora of rats are affected considerably by high-altitude hypoxic environments [72].

Although considerable progress has been made in preventing and treating high-altitude illnesses, studies on the rational clinical use and efficacy of drugs at high altitudes are limited. In summary, the findings of the present study showed that the pharmacokinetics of nifedipine, bosentan, simvastatin, and sildenafil in rats, including MRT, t_1/2z_, CL_z_/F, and V_z_/F, were significantly affected by exposure to high-altitude hypoxia. Pharmacokinetic studies in humans are necessary to determine the optimal dosage of nifedipine, bosentan, simvastatin, and sildenafil in high-altitude environments. Moreover, the effects of high-altitude hypoxia on the systemic pharmacokinetics of other drugs require further clinical evaluation.

## 4. Materials and Methods

### 4.1. Reagents and Instruments

All chemicals and solvents used were of the highest available grade. Nifedipine, sildenafil, bosentan, simvastatin, and losartan were obtained from Macklin Biochemical Co., Ltd. (lot numbers: C10195240, C11374655, C10084046, C10636115, C10103850; Shanghai, China). Nimodipine, lovastatin, N-desmethyl sildenafil, hydroxybosentan, and simvastatin hydroxy acid were obtained from J&K Scientific Co., Ltd. (lot numbers: L780R03, LTB0S49, LE90U01, 1YHH522, H830270; Beijing, China). Dehydronifedipine, diazepam, and lovastatin hydroxy acid were obtained from Alta Scientific Co., Ltd. (lot numbers: 1ST10041, 1ST7102, S065495; Tianjin, China). The SDS-PAGE gel preparation kit was obtained from Soleibao Technology Co., Ltd. (lot number: 20211013; Beijing, China). The Rainbow 245 Broad Spectrum Protein Marker was obtained from Soleibo Technology Co., Ltd. (lot number: 20210416; Beijing, China). The anti-β-actin antibody was obtained from Boaosen Biotechnology Co., Ltd. (lot number: AH11286487; Beijing, China). The goat anti-rabbit IgG-HRP secondary antibody was obtained from Aibotech Biotechnology Co., Ltd. (lot number: 9300014001; Wuhan, China). FastPure Cell/Tissue Total RNA Isolation Kit, HiScript® II Q RT SuperMix for qPCR Kit, and ChamQ Universal SYBR qPCR Master Mix Kit were purchased from Novozymes Technology Co., Ltd. (lot numbers: 7E570E1, 7E242E8, 7E340F9; China). Primers used for qRT-PCR were synthesized by Sangon Biotech (Shanghai, China).

### 4.2. Animals and Experimental Treatments

Male Sprague Dawley rats weighing 180–220 g were obtained from the Laboratory Animal Center of Xi’an Jiaotong University Medical College (certificate No.: 2018-001). All rats were housed per cage in separate rooms at a constant temperature (22 ± 2 °C) and humidity (55 ± 10%) under a 12 h light/12 h dark cycle. The rats were allowed to adapt for a week and had ad libitum access to water and food pellets. All experimental procedures were performed in strict accordance with the National Institutes of Health Guide for the Care and Use of Laboratory Animals (Approval No.: 2017-15).

After the adaptation period, 32 male rats were randomly divided into normoxia (N, 390 m, PaO_2_:20.2 kPa), acute hypoxia (AH, 4300 m; PaO_2_, 12.5 kPa), chronic hypoxia (CH, 4300 m; PaO_2_, 12.5 kPa), and chronic hypoxia to normoxia (CH-N, 390 and 4300 m; PaO_2_: 20.2 and 12.5 kPa) groups, with eight rats per group. Rats in the normoxia group lived at an altitude of approximately 390 m in Xi’an, Shanxi Province, China. Rats in the acute hypoxia and chronic hypoxia groups received a 72 h acute hypoxia exposure and a 30-day chronic hypoxia exposure, respectively, at an altitude of 4300 m and oxygen partial pressure of 11.1 kPa in Maduo, Qinghai Province. The rats in these two groups were airlifted from Xi’an to Xining City by plane and then transported by bus to Maduo County, the whole trip was completed in 10 h. Rats in the chronic hypoxia to normoxia group were returned to Xi’an and raised for 7 days after a 30-day chronic hypoxia exposure in Maduo. The rats in the chronic hypoxia to normoxia group were transported by bus from Maduo to Xi’an; the whole trip was completed in 24 h. 

Sample collection of the normoxia and chronic hypoxia to normoxia groups was carried out at Xi’an Jiaotong University Medical College, whereas the sample collection of the acute hypoxia and chronic hypoxia groups was performed at Maduo County Hospital. Hematological and serum biochemical parameters were performed immediately after sampling, and the tissues for Hematoxylin-Eosin staining were fixed immediately. The time from tissue sampling to putting the tissue into a paraformaldehyde solution was up to 1 minute. To determine the protein and mRNA expression of CYP3A1 in rat liver, all liver samples were stored in liquid nitrogen, and the time from liver sampling to putting the liver into liquid nitrogen was up to 1 minute. The liver samples were transported to the Research Center for High Altitude Medicine of Qinghai University by bus. To determine the pharmacokinetics of the drugs, all plasma samples were stored in liquid nitrogen and were transported to the State Key Laboratory of Plateau Ecology and Agriculture of Qinghai University by bus.

### 4.3. Determination of Physiologic and Biochemical Parameters

Rats in each group were anesthetized via intraperitoneal injection of 20% urethane before blood sampling. From the ophthalmic venous plexus, 0.3 mL of whole blood was collected in anticoagulant tubes containing EDTA-2Na. Routine blood examinations, including white blood cells, hemoglobin, red blood cells, hematocrit, mean corpuscular volume, platelet count, and mean platelet volume, were performed to determine the response of the rats to high-altitude conditions using an XN-10 automatic hematology analyzer (Sysmex Corporation, Kobe, Japan).

From the ophthalmic venous plexus, 0.5 ml of whole blood was collected in the tube without anticoagulant and centrifuged, and the serum was extracted. The blood biochemical parameters of rats exposed to high-altitude hypoxia were examined, including alanine aminotransferase (ALT), aspartate transaminase (AST), total protein, albumin, globulin, urea, and glucose, using an AU5800 automatic biochemistry analyzer (Olympus Corporation, Tokyo, Japan).

### 4.4. Hematoxylin-Eosin Staining

Rats in each group were anesthetized using an intraperitoneal injection of 20% urethane, and the heart, lungs, liver, and kidney tissues of the rats in each group were carefully sampled to avoid mechanical damage and extrusion during the process. Each tissue was dehydrated, embedded in wax, sliced into 5 µm sections using a tissue slicer (Hangu medical technology, Hangzhou, China), baked at 60 °C for 30 min, stained with hematoxylin-eosin, and sealed with neutral gum. The sections were observed under a microscope (Nikon Corporation, Tokyo, Japan) and photographed.

### 4.5. Pharmacokinetic Study Design

To evaluate the effect of high-altitude hypoxia on the pharmacokinetics of simvastatin and its main metabolite, simvastatin hydroxy acid, rats in the normoxia, acute hypoxia, chronic hypoxia, and chronic hypoxia to normoxia groups were orally administered a simvastatin solution at a dose of 10.0 mg/kg after an overnight fast of not less than 12 h (with water allowed ad libitum). Serial blood samples (0.3 mL) were collected from the eye socket before (baseline) and at 10, 15, and 30 min and 1, 2, 4, 6, 8, 10, 12, and 24 h after drug administration. To evaluate the effect of high-altitude hypoxia on the pharmacokinetics of sildenafil, bosentan, nifedipine, and their main metabolites, N-desmethyl sildenafil, hydroxybosentan, and dehydronifedipine, respectively, rats in the normoxia, acute and chronic hypoxia, and chronic hypoxia to normoxia groups were orally administered sildenafil, bosentan, and nifedipine solution at doses of 10.0, 20.0, 2.0 mg/kg, respectively, after an overnight fast of not less than 12 h (with water allowed ad libitum). To obtain the plasma of rats medicated with sildenafil and N-desmethyl sildenafil, 0.3 mL of serial blood samples was collected from the eye socket before (baseline) and 10, 30, 45 min, 1, 1.5, 2, 4, 6, 8, 12, and 24 h after drug administration. To obtain the plasma of rats medicated with bosentan and hydroxybosentan, 0.3 mL of serial blood samples was collected from the eye socket before (baseline) and 10 and 30 min and 1, 2, 3, 4, 6, 8, 10, 12, and 24 h after drug administration. To obtain the plasma of rats medicated with nifedipine and dehydronifedipine, 0.3 mL of serial blood samples was collected from the eye socket before (baseline) and 10, 20, 30, and 45 min and 1, 1.5, 2, 4, 6, 8, and 12 h after drug administration. All blood samples were centrifuged at 1000× *g* for 15 min at 4 °C, and the plasma was separated and immediately stored at −20 °C for further analyses.

### 4.6. Sample Processing

The plasma sample was processed with acetonitrile to denature the protein, and a mixture of acetonitrile, internal standard (IS), and plasma was vortexed for 1 min and then centrifuged at 13,000× *g* for 10 min at 4 °C. An aliquot of the supernatant was analyzed using ultra-high-performance liquid chromatography-mass spectrometry (UHPLC-MS) (Thermo Scientific Ultimate 3000 and Q-Exactive Focus MS, Sunnyvale, CA, USA).

### 4.7. Chromatographic Conditions

An HPLC separation module (Thermo Scientific UltiMate 3000) with a cooled autosampler and a column oven was used for this assay. A Thermo Hypersil GOLD C18 column (50 mm × 2.1 mm, 1.9 µm; Thermo Fisher) was used for chromatographic separation. The gradient elution was performed as follows:

Simvastatin and simvastatin hydroxy acid: injection volume, 4 μL; internal standard, lovastatin and lovastatin hydroxy acid; flow rate, 0.3 mL/min; column temperature, 35 °C; Linear gradient elution procedure is shown in Table 9.

Sildenafil and N-desmethyl sildenafil: injection volume, 3 μL; internal standard, diazepam; flow rate, 0.3 mL/min; column temperature, 35 °C; Linear gradient elution procedure is shown in Table 10.

Nifedipine and dehydronifedipine: injection volume, 8 μL; internal standard, nimodipine; flow rate, 0.3 mL/min; column temperature, 35 °C; Linear gradient elution procedure is shown in Table 11.

Bosentan and hydroxybosentan: injection volume, 8 μL; internal standard, losartan; flow rate, 0.3 mL/min; column temperature, 35 °C; Linear gradient elution procedure is shown in Table 12.

### 4.8. Mass Spectrometry Conditions

A high-resolution mass spectrum (Q-Exactive Focus MS) was used to analyze samples in positive electrospray ionization mode, except simvastatin and simvastatin hydroxy acid (positive and negative electrospray ionization mode), and the data were collected and processed using Trace Finder 4.0. The adopted source parameters were as follows: sheath gas flow rate, 35 arb; aux gas flow rate, 10 arb; sweep gas flow rate, 1 arb; spray voltage, 3.5 kV; capillary temp, 260 °C; s-lens RF level, 50%; aux gas heater temperature, 300 °C. The m/z were as follows: simvastatin, 441.261; simvastatin hydroxy acid, 435.2752; lovastatin (IS), 427.2455; lovastatin hydroxy acid (IS), 421.2595; sildenafil, 475.2122; N-desmethyl sildenafil, 461.1966; diazepam (IS), 285.0789; nifedipine, 347.12376; dehydronifedipine, 345.10811; nimodipine (IS), 419.18128; bosentan, 552.1911; hydroxybosentan, 568.1860; and losartan (IS), 461.1253.

### 4.9. UHPLC-MS Validation

The established UHPLC-MS method used in this study was validated according to the 2020 edition of the Chinese Pharmacopoeia guidelines and the relevant guidelines of the US FDA. The following aspects were investigated: specificity, standard curve performance, accuracy, precision, recovery, and stability. Both precision and accuracy were measured using four concentrations of quality control samples; five samples were analyzed for each concentration, and three analysis batches were measured consecutively. The recovery rate refers to the extraction recovery rate, and each compound was analyzed at three concentrations. The stability of the quality control samples of each compound was investigated under the following conditions: at 25 °C for 8 h, placed in an automatic sample injection tray at 20 °C for 24 h, and freezing and thawing at −20 °C three times. Plasma samples were only tested after each indicator was verified to meet the requirements.

### 4.10. Pharmacokinetic Analysis

The pharmacokinetic values of nifedipine, bosentan, simvastatin, sildenafil, and their respective metabolites, dehydronifedipine, hydroxybosentan, simvastatin hydroxy acid, and N-desmethyl sildenafil, were calculated for each rat, and the mean values were then determined for analysis. The area under the concentration time curve (AUC), AUC of the first moment (AUMC), mean residence time (MRT), half-life (t_1/2z_), total plasma clearance (CL_z_/F), apparent volume of distribution (V_z_/F), and elimination rate constant (λ_z_) were calculated under a noncompartmental analysis mode using DAS 2.0 software (Institute of Clinical Pharmacology, Shanghai University of Traditional Chinese Medicine, China). The peak time (T_max_) and maximum plasma concentration (C_max_) were obtained directly from the original data.

### 4.11. Western Blot

The liver tissues of rats in each group were sampled to evaluate the effect of high-altitude hypoxia on the protein expression levels of CYP3A1. Total protein samples were harvested from liver tissue and homogenized, followed by protein content determination. The protein samples were separated by electrophoresis on 10% SDS polyacrylamide gels and then transferred to a polyvinylidene fluoride membrane. Tris-buffered saline Tween-20 was used to block membranes with 5% skimmed milk powder for 1.5 h. The polyvinylidene fluoride membranes were incubated with primary antibodies against β-actin (1:1000, mouse monoclonal, Abcam, ab8226) and CYP3A1 (1:1000, rabbit monoclonal, Abcam, ab22733) at 4 °C overnight. The membranes were incubated with a 1:10000 dilution of the secondary antibody at 25 °C for 50 min. Specific protein bands were detected using an ECL system (Amersham Imager 600, Boston, MA, USA).

### 4.12. Quantitative Real-Time Polymerase Chain Reaction

The livers of rats from each group were excised immediately after death and stored at –80 °C before use. Approximately 100–200 mg of liver tissue was homogenized, and total RNA was isolated using a Total RNA Isolation Kit from Vazyme Biotech Co., Ltd. (7E570E1, Nanjing, China). RNA solution quality was assessed using a NanoDrop 2000c spectrophotometer (Thermo Fisher Scientific, Waltham, MA, USA), and cDNA was synthesized using a HiScript II Q RT SuperMix for qPCR from Vazyme Biotech Co., Ltd. (7E242E8, Nanjing, China), according to the manufacturer’s instructions. The relative mRNA expression of CYP3A1 was analyzed using a Bio-Rad CFX Connect Real-Time PCR System (Bio-Rad Laboratories, Inc., Hercules, CA, USA). PCR products were amplified at 95 °C for 40 s, followed by 40 cycles at 60 °C for 30 s, 95 °C for 15 s, 60 °C for 60 s, and 95 °C for 1 s. The fold-induction values were calculated using the 2^−ΔΔCt^ method, where ΔCt represents the difference in cycle threshold numbers between the target gene and the control gene and ΔΔCt represents the relative change in the difference between the control and treatment groups. qPCR efficiency for all samples was calculated according to the Minimum Information for Publication of Quantitative Real-Time PCR Experiments guidelines.

CYP3A1 primer sequence:

5′ - CGTTCACCAGTGGAAGACTCAAGG - 3′ (forward primer);

5′ - TTCTTTCACAGGGACAGGTTTGCC - 3′ (reverse primer).

β-actin primer sequence:

5′ - TCACCAACTGGGACGATATG - 3′ (forward primer);

5′ - GTTGGCCTTAGGGTTCAGAG - 3′ (reverse primer).

### 4.13. Statistical Analysis

All numerical data were expressed as the mean ± standard deviation (SD). Statistical differences were determined using one-way analysis of variance (ANOVA). Comparisons between two groups were performed using the least significant difference (LSD). The difference of *p* < 0.05 was considered statistically significant. The Dunnett’s t test was used for variance nonhomogeneity. All the statistical analyses were performed with Statistical Package for the Social Sciences, version 20.0 (SPSS Inc., Chicago, IL, USA).

## Figures and Tables

**Figure 1 pharmaceuticals-15-01303-f001:**
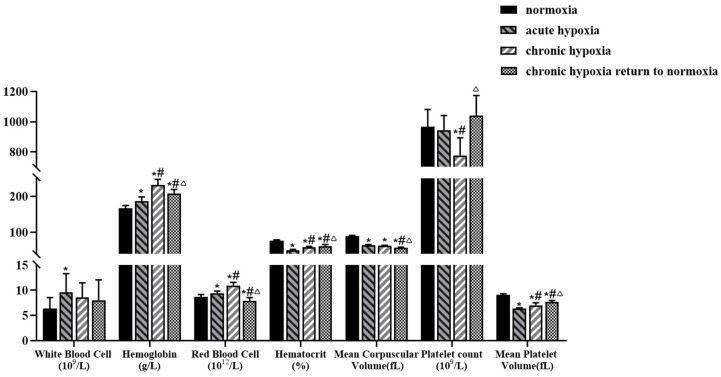
Changes in white blood cell, hemoglobin, red blood cell, hematocrit, mean corpuscular volume, platelet count, and mean platelet volume in rats after exposure to high-altitude hypoxia. The data are presented as mean ± SD. *n* = 8. * *p* < 0.05 compared to the normoxia group; # *p* < 0.05 compared to the acute hypoxia group; Δ *p* < 0.05 compared to the chronic hypoxia group.

**Figure 2 pharmaceuticals-15-01303-f002:**
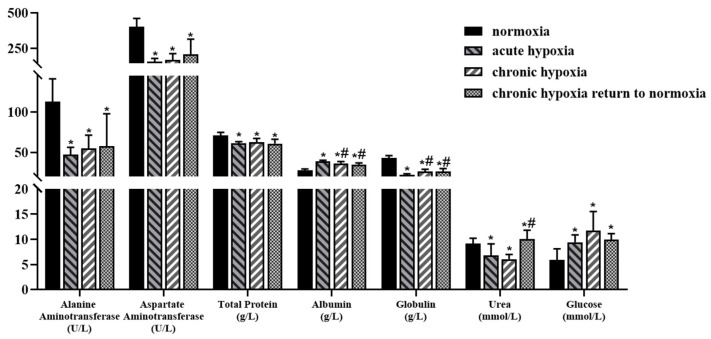
Changes in alanine aminotransferase, aspartate aminotransferase, total protein, albumin, globulin, urea, and glucose in rats after exposure to high-altitude hypoxia. The data are presented as mean ± SD. *n* = 8. * *p* < 0.05 compared to the normoxia group; # *p* < 0.05 compared to the acute hypoxia group.

**Figure 3 pharmaceuticals-15-01303-f003:**
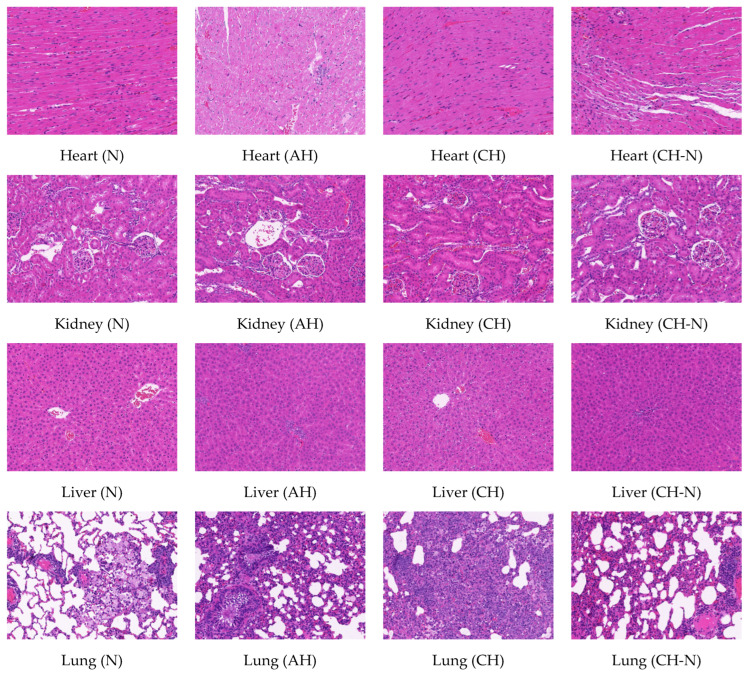
Effects of high-altitude hypoxia on heart, kidney, liver, and lung tissue in rats (HE, 200×). Note: N, AH, CH, CH-N refer to the normoxia group, acute hypoxia group, chronic hypoxia group, and chronic hypoxia to normoxia group, respectively.

**Figure 4 pharmaceuticals-15-01303-f004:**
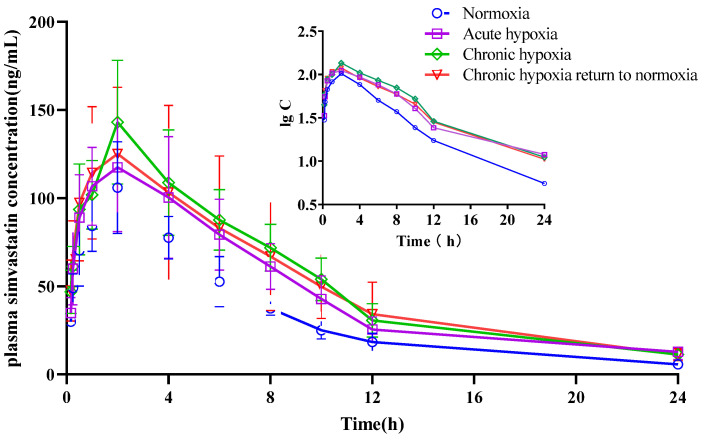
Mean plasma concentration time curve of simvastatin for rats after exposure to high-altitude hypoxia (*n* = 8).

**Figure 5 pharmaceuticals-15-01303-f005:**
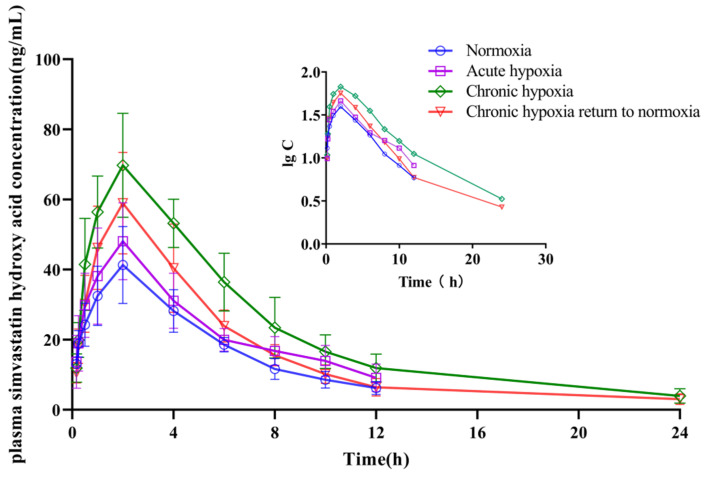
Mean plasma concentration time curve of simvastatin hydroxy acid for rats after exposure to high-altitude hypoxia (*n* = 8).

**Figure 6 pharmaceuticals-15-01303-f006:**
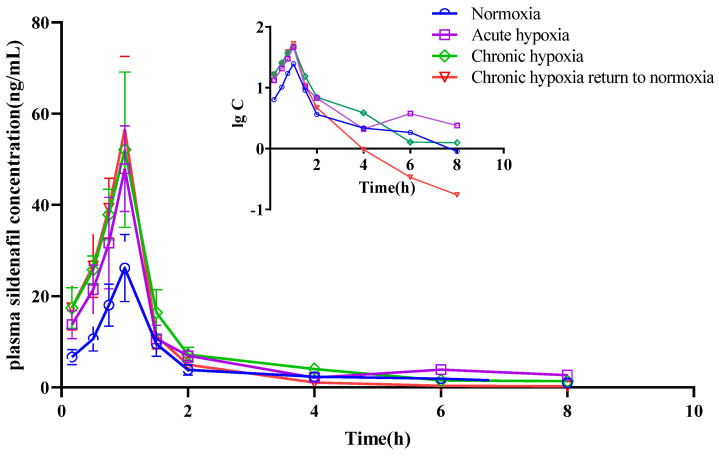
Mean plasma concentration time curve of sildenafil for rats after exposure to high-altitude hypoxia (*n* = 8).

**Figure 7 pharmaceuticals-15-01303-f007:**
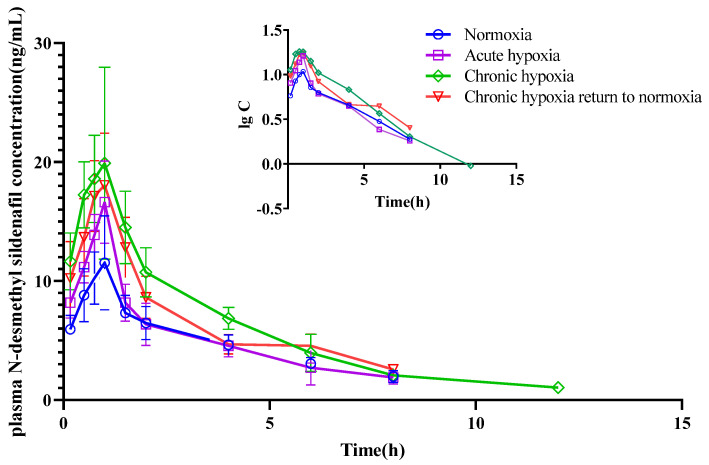
Mean plasma concentration time curve of N-desmethyl sildenafil for rats after exposure to high-altitude hypoxia (*n* = 8).

**Figure 8 pharmaceuticals-15-01303-f008:**
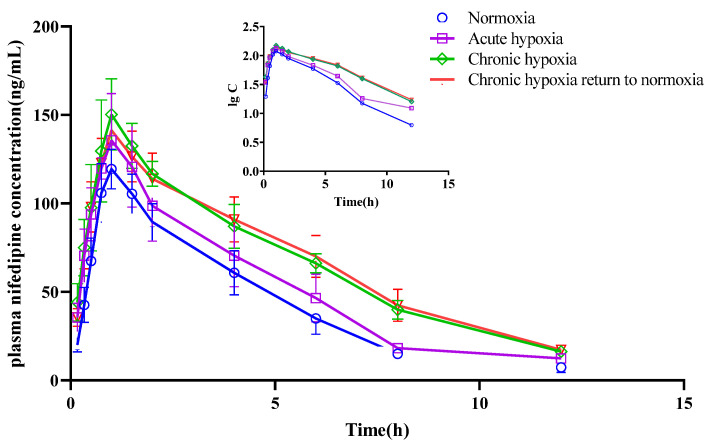
Mean plasma concentration time curve of nifedipine for rats after exposure to high-altitude hypoxia (*n* = 8).

**Figure 9 pharmaceuticals-15-01303-f009:**
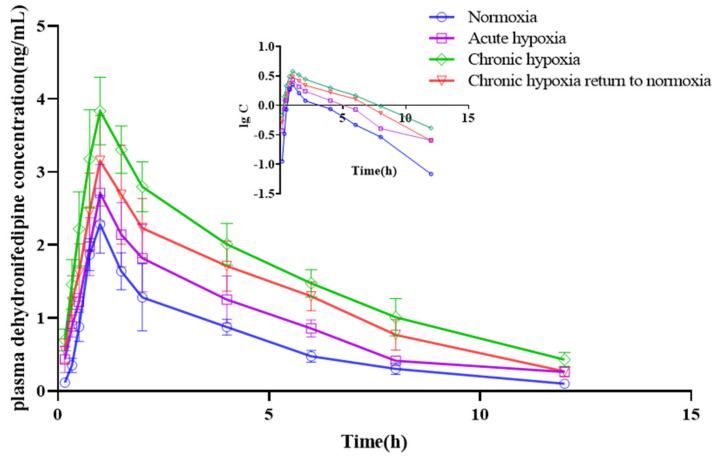
Mean plasma concentration time curve of dehydronifedipine for rats after exposure to high-altitude hypoxia (*n* = 8).

**Figure 10 pharmaceuticals-15-01303-f010:**
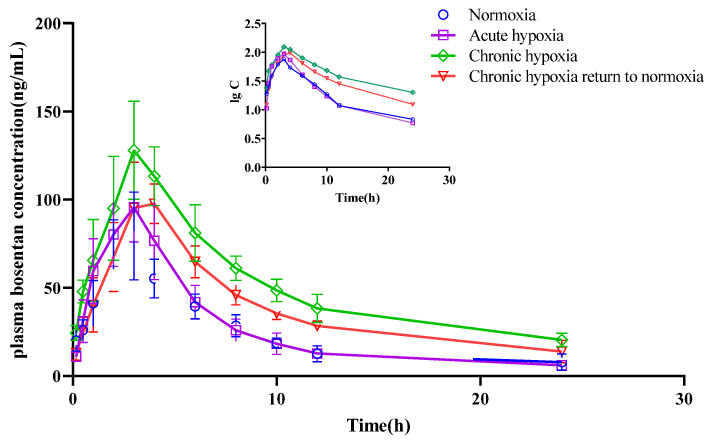
Mean plasma concentration time curve of bosentan for rats after exposure to high-altitude hypoxia (*n* = 8).

**Figure 11 pharmaceuticals-15-01303-f011:**
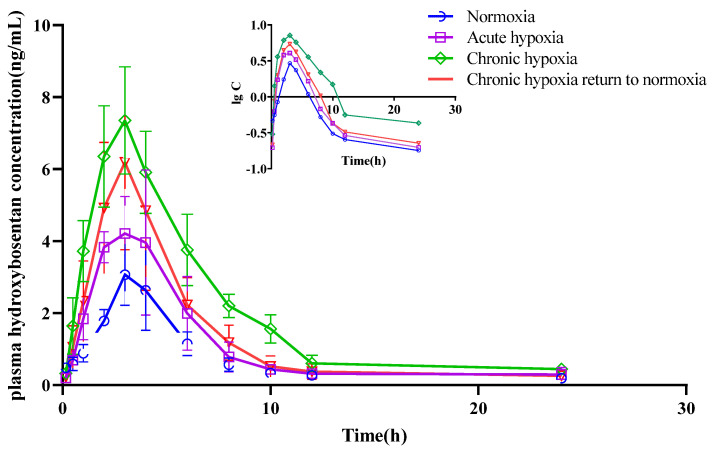
Mean plasma concentration time curve of hydroxybosentan for rats after exposure to high-altitude hypoxia (*n* = 8).

**Figure 12 pharmaceuticals-15-01303-f012:**
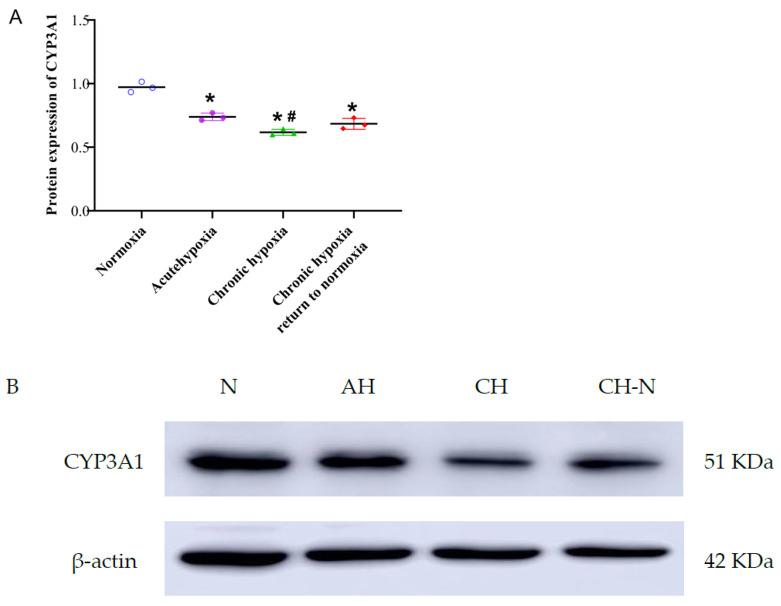
The protein expression of CYP3A1 in rats after exposure to high-altitude hypoxia (**A**). Liver protein expression map of four groups of rats (**B**). N: normoxia group; AH: acute hypoxia group; CH: chronic hypoxia group; CH-N: chronic hypoxia to normoxia group. The data are presented as the mean ± SD. *n* = 3. * *p* < 0.05 compared to the normoxia group, ^#^
*p* < 0.05 compared to the acute group.

**Figure 13 pharmaceuticals-15-01303-f013:**
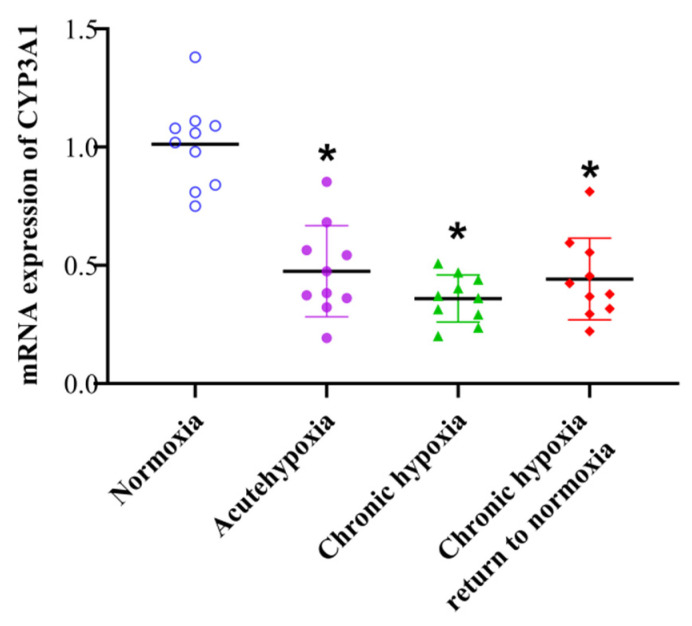
The mRNA expression of CYP3A1 in rats after exposure to high-altitude hypoxia. The data are presented as the mean ± SD. *n* = 10. * *p* < 0.05 compared to the normoxia group.

**Table 1 pharmaceuticals-15-01303-t001:** Pharmacokinetic parameters of simvastatin in rat plasma after exposure to high-altitude hypoxia.

Parameters	Normoxia	Acute Hypoxia	Chronic Hypoxia	Chronic Hypoxia to Normoxia
AUC_(0–24)_, h·ug/L	810.05 ± 146.22	1127.94 ± 317.36	1268.96 ± 177.10	1239.53 ± 566.75
AUC_(0–∞)_, h·ug/L	842.00 ± 148.01	1191.85 ± 334.85	1377.89 ± 199.13 ^a^	1317.55 ± 594.18
AUMC_(0–24)_, h^2^·ug/L	5124.82 ± 992.08	8041.01 ± 2385.49	8895.72 ± 1463.89 ^a^	8854.69 ± 4261.74 ^a^
AUMC_(0–∞)_, h^2^·ug/L	6419.60 ± 1258.50	11,267.41 ± 3647.81 ^a^	12,561.54 ± 2533.47 ^a^	11,797.45 ± 5201.98 ^a^
MRT_(0–24)_, h	6.32 ± 0.60	7.15 ± 0.86 ^a^	7.00 ± 0.54	7.08 ± 0.70 ^a^
MRT_(0–∞)_, h	7.63 ± 0.89	9.53 ± 2.06 ^a^	9.07 ± 0.98 ^a^	9.02 ± 1.26
t_1/2z_, h	4.99 ± 1.16	5.37 ± 1.11	6.55 ± 0.92 ^ab^	5.61 ± 0.93
T_max_, h	1.75 ± 0.46	1.44 ± 0.62	2.50 ± 0.93 ^b^	2.25 ± 1.16
CL_z_/F, L/h/kg	12.29 ± 2.73	8.90 ± 2.14 ^a^	7.42 ± 1.33 ^a^	9.04 ± 3.91 ^a^
V_z_/F, L/kg	89.22 ± 29.43	68.84 ± 19.49	69.76 ± 12.46	72.80 ± 29.95
λ_z_, 1/h	0.15 ± 0.05	0.13 ± 0.03	0.11 ± 0.02 ^a^	0.13 ± 0.02
C_max_, ug/L	107.82 ± 30.16	133.29 ± 34.51	156.23 ± 26.82 ^a^	134.74 ± 50.76

Note: Data are expressed as the mean ± SD values (*n* = 8). ^a^
*p* < 0.05 compared to the normoxia group; ^b^
*p* < 0.05 compared to the acute hypoxia group. AUC, area under the concentration time curve; AUMC, AUC of the first moment; MRT, mean residence time; t_1/2z_, half-life; T_max_, peak time; CL_z_/F, total plasma clearance; V_z_/F, apparent volume of distribution; λ_z_, elimination rate constant; C_max_, the maximum plasma concentration.

**Table 2 pharmaceuticals-15-01303-t002:** Pharmacokinetic parameters of simvastatin hydroxy acid in rat plasma after exposure to high-altitude hypoxia.

Parameters	Normoxia	Acute Hypoxia	Chronic Hypoxia	Chronic Hypoxia to Normoxia
AUC_(0–24)_, h·ug/L	242.33 ± 55.06	289.41 ± 77.21	533.64 ± 76.15 ^ab^	382.52 ± 62.29 ^abc^
AUC_(0–∞)_, h·ug/L	284.51 ± 63.54	364.93 ± 113.48	560.05 ± 84.85 ^ab^	399.30 ± 64.67 ^ac^
AUMC_(0–24)_, h^2^·ug/L	1034.71 ± 226.74	1325.11 ± 374.48	3398.47 ± 734.33 ^ab^	2262.16 ± 552.26 ^abc^
AUMC_(0–∞)_, h^2^·ug/L	1860.09 ± 554.91	2635.37 ± 1238.53	4377.03 ± 1158.46 ^ab^	3043.17 ± 1072.49 ^ac^
MRT_(0–24)_, h	4.27 ± 0.31	4.58 ± 0.30	6.33 ± 0.63 ^ab^	5.91 ± 1.10 ^ab^
MRT_(0–∞)_, h	6.55 ± 1.29	7.03 ± 1.05	7.75 ± 1.06	7.60 ± 2.34
t_1/2z_, h	4.68 ± 1.27	5.00 ± 0.82	5.44 ± 1.54	5.22 ± 1.37
T_max_, h	2.00 ± 0.00	1.88 ± 0.35	2.12 ± 0.83	2.12 ± 0.84
CL_z_/F, L/h/kg	36.69 ± 7.97	30.62 ± 13.14	18.22 ± 2.82 ^ab^	25.53 ± 3.44 ^a^
V_z_/F, L/kg	249.82 ± 97.60	213.18 ± 70.26	143.73 ± 47.89 ^a^	190.51 ± 52.21
λ_z_, 1/h	0.16 ± 0.04	0.14 ± 0.02	0.14 ± 0.05	0.15 ± 0.06
C_max_, ug/L	41.29 ± 13.14	50.38 ± 14.76	73.94 ± 10.53 ^ab^	63.20 ± 17.89 ^a^

Note: Data are expressed as the mean ± SD values (*n* = 8). ^a^
*p* < 0.05 compared to the normoxia group; ^b^
*p* < 0.05 compared to the acute hypoxia group; ^c^
*p* < 0.05 compared to the chronic hypoxia group.

**Table 3 pharmaceuticals-15-01303-t003:** Pharmacokinetic parameters of sildenafil in rat plasma after exposure to high-altitude hypoxia.

Parameters	Normoxia	Acute Hypoxia	Chronic Hypoxia	Chronic Hypoxia to Normoxia
AUC_(0–24)_, h·ug/L	37.46 ± 6.68	63.56 ± 8.71 ^a^	73.62 ± 13.42 ^a^	65.40 ± 13.34 ^a^
AUC_(0–∞)_, h·ug/L	42.88 ± 7.86	76.85 ± 8.14 ^a^	80.61 ± 17.50 ^a^	73.91 ± 16.71 ^a^
AUMC_(0–24)_, h^2^·ug/L	81.84 ± 16.63	141.81 ± 27.98 ^a^	175.85 ± 46.60 ^a^	120.72 ± 34.95 ^c^
AUMC_(0–∞)_, h^2^·ug/L	145.62 ± 47.35	337.77 ± 121.66 ^a^	310.52 ± 171.71 ^a^	261.06 ± 179.05
MRT_(0–24)_, h	2.19 ± 0.27	2.23 ± 0.32	2.37 ± 0.33	1.85 ± 0.38 ^abc^
MRT_(0–∞)_, h	3.42 ± 1.04	4.36 ± 1.35	3.73 ± 1.31	3.39 ± 1.52
t_1/2z_, h	3.22 ± 1.06	3.44 ± 1.03	3.69 ± 0.88	3.60 ± 1.85
T_max_, h	0.94 ± 0.12	0.97 ± 0.09	0.94 ± 0.12	0.97 ± 0.09
CL_z_/F, L/h/kg	240.34 ± 45.04	13,150 ± 14.78 ^a^	129.83 ± 31.14 ^a^	141.55 ± 32.28 ^a^
V_z_/F, L/kg	1096.51 ± 346.56	651.80 ± 190.65 ^a^	677.15 ± 164.99 ^a^	722.91 ± 368.21 ^a^
λ_z_, 1/h	0.23 ± 0.07	0.22 ± 0.08	0.20 ± 0.05	0.23 ± 0.09
C_max_, ug/L	27.64 ± 6.70	49.89 ± 8.28 ^a^	58.03 ± 10.66 ^a^	58.06 ± 16.10 ^a^

Note: Data are expressed as the mean ± SD values (*n* = 8). ^a^
*p* < 0.05 compared to the normoxia group; ^b^
*p* < 0.05 compared to the acute hypoxia group; ^c^
*p* < 0.05 compared to the chronic hypoxia group.

**Table 4 pharmaceuticals-15-01303-t004:** Pharmacokinetic parameters of N-desmethyl sildenafil in rat plasma after exposure to high-altitude hypoxia.

Parameters	Normoxia	Acute Hypoxia	Chronic Hypoxia	Chronic Hypoxia to Normoxia
AUC_(0–24)_, h·ug/L	39.30 ± 7.26	42.36 ± 9.23	71.30 ± 8.60 ^ab^	54.45 ± 9.51 ^abc^
AUC_(0–∞)_, h·ug/L	50.11 ± 9.99	53.03 ± 15.10	80.34 ± 12.41 ^ab^	69.36 ± 11.86 ^ab^
AUMC_(0–24)_, h^2^·ug/L	119.08 ± 18.82	117.10 ± 36.76	254.22 ± 39.18 ^ab^	156.50 ± 28.71 ^abc^
AUMC_(0–∞)_, h^2^·ug/L	264.43 ± 91.40	270.03 ± 160.34	462.33 ± 157.79 ^ab^	374.28 ± 169.66
MRT_(0–24)_, h	3.05 ± 0.22	2.74 ± 0.27 ^a^	3.56 ± 0.28 ^ab^	2.88 ± 0.13 ^c^
MRT_(0–∞)_, h	5.19 ± 1.07	4.88 ± 1.30	5.64 ± 1.22	5.29 ± 1.77
t_1/2z_, h	3.56 ± 0.92	3.67 ± 1.12	3.66 ± 1.21 ^b^	3.81 ± 1.62
T_max_, h	0.91 ± 0.13	0.94 ± 0.12	0.81 ± 0.22	0.97 ± 0.28
CL_z_/F, L/h/kg	207.02 ± 43.32	198.12 ± 39.05	127.05 ± 19.20 ^ab^	148.14 ± 26.80 ^ab^
V_z_/F, L/kg	1038.91 ± 239.58	1019.96 ± 262.29	646.96 ± 159.90 ^ab^	793.51 ± 280.15
λ_z_, 1/h	0.21 ± 0.07	0.20 ± 0.06	0.21 ± 0.08	0.20 ± 0.06
C_max_, ug/L	12.08 ± 4.34	17.16 ± 3.82	22.12 ± 7.83 ^a^	19.61 ± 4.03 ^a^

Note: Data are expressed as the mean ± SD values (*n* = 8). ^a^
*p* < 0.05 compared to the normoxia group; ^b^
*p* < 0.05 compared to the acute hypoxia group; ^c^
*p* < 0.05 compared to the chronic hypoxia group.

**Table 5 pharmaceuticals-15-01303-t005:** Pharmacokinetic parameters of nifedipine in rat plasma after exposure to high-altitude hypoxia.

Parameters	Normoxia	Acute Hypoxia	Chronic Hypoxia	Chronic Hypoxia to Normoxia
AUC_(0–24)_, h·ug/L	511.92 ± 56.80	611.57 ± 75.70 ^a^	799.00 ± 60.34 ^ab^	811.72 ± 71.18 ^ab^
AUC_(0–∞)_, h·ug/L	544.41 ± 67.37	664.25 ± 69.52 ^a^	881.48 ± 71.37 ^ab^	896.95±77.35 ^ab^
AUMC_(0–24)_, h^2^·ug/L	1820.82 ± 279.73	2247.74 ± 286.46 ^a^	3328.84 ± 277.18 ^ab^	3471.50 ± 395.30 ^ab^
AUMC_(0–∞)_, h^2^·ug/L	2315.83 ± 514.25	3173.46 ± 320.66 ^a^	4719.59 ± 674.31 ^ab^	4904.58 ± 493.11 ^ab^
MRT_(0–24)_, h	3.54 ± 0.24	3.68 ± 0.26	4.17 ± 0.18 ^ab^	4.27 ± 0.21 ^ab^
MRT_(0–∞)_, h	4.21 ± 0.56	4.81 ± 0.57 ^a^	5.34 ± 0.51 ^ab^	5.46 ± 0.20 ^ab^
t_1/2z_, h	2.75 ± 0.47	3.29 ± 0.55 ^a^	3.41 ± 0.48 ^a^	3.40 ± 0.29 ^a^
T_max_, h	0.94 ± 0.17	1.06 ± 0.29	1.09 ± 0.26	1.03 ± 0.21
CL_z_/F, L/h/kg	3.73 ± 0.50	3.04 ± 0.28 ^a^	2.28 ± 0.175 ^ab^	2.24 ± 0.20 ^ab^
V_z_/F, L/kg	14.64 ± 2.30	14.52 ± 3.36	11.21 ± 1.67 ^ab^	10.98 ± 1.19 ^ab^
λ_z_, 1/h	0.26 ± 0.05	0.22 ± 0.03 ^a^	0.21 ± 0.03 ^a^	0.20 ± 0.02 ^a^
C_max_, ug/L	122.36 ± 11.87	145.16 ± 25.74	162.14 ± 20.13 ^a^	143.50 ± 11.51

Note: Data are expressed as the mean ± SD values (*n* = 8). ^a^
*p* < 0.05 compared to the normoxia group; ^b^
*p* < 0.05 compared to the acute hypoxia group.

**Table 6 pharmaceuticals-15-01303-t006:** Pharmacokinetic parameters of dehydronifedipine in rat plasma after exposure to high-altitude hypoxia.

Parameters	Normoxia	Acute Hypoxia	Chronic Hypoxia	Chronic Hypoxia to Normoxia
AUC_(0–24)_, h·ug/L	7.72 ± 0.71	11.17 ± 1.26 ^a^	19.06 ± 2.46 ^ab^	15.43 ± 2.45 ^abc^
AUC_(0–∞)_, h·ug/L	8.20 ± 0.79	12.19 ± 1.18 ^a^	21.42 ± 3.14 ^ab^	17.17 ± 3.58 ^abc^
AUMC_(0–24)_, h^2^·ug/L	27.29 ± 3.17	42.90 ± 5.49 ^a^	80.12 ± 11.97 ^ab^	63.61 ± 10.37 ^abc^
AUMC_(0–∞)_, h^2^·ug/L	34.01 ± 8.78	61.65 ± 10.54 ^a^	119.36 ± 24.90 ^ab^	87.16 ± 22.26 ^abc^
MRT_(0–24)_, h	3.54 ± 0.30	3.84 ± 0.21 ^a^	4.20 ± 0.15 ^ab^	4.14 ± 0.29 ^ab^
MRT_(0–∞)_, h	4.13 ± 0.89	5.05 ± 0.71 ^a^	5.54 ± 0.58 ^a^	5.09 ± 0.73 ^a^
t_1/2z_, h	2.68 ± 0.95	3.22 ± 0.80	3.54 ± 0.57	3.25 ± 1.16
T_max_, h	0.91 ± 0.13	1.16 ± 0.30 ^a^	1.03 ± 0.21	1.03 ± 0.21
CL_z_/F, L/h/kg	245.99 ± 25.34	165.37 ± 15.95 ^a^	95.13 ± 13.88 ^ab^	120.46 ± 22.30 ^abc^
V_z_/F, L/kg	938.91 ± 301.42	772.88 ± 227.86	480.24 ± 75.72 ^a^	553.99 ± 190.27 ^a^
λ_z_, 1/h	0.29 ± 0.11	0.23 ± 0.06	0.20 ± 0.04 ^a^	0.24 ± 0.10
C_max_, ug/L	2.31 ± 0.46	2.91 ± 0.27	3.98 ± 0.37 ^ab^	3.20 ± 0.76 ^ac^

Note: Data are expressed as the mean ± SD values (*n* = 8). ^a^
*p* < 0.05 compared to the normoxia group; ^b^
*p* < 0.05 compared to the acute hypoxia group; ^c^
*p* < 0.05 compared to the chronic hypoxia group.

**Table 7 pharmaceuticals-15-01303-t007:** Pharmacokinetic parameters of bosentan in rat plasma after exposure to high-altitude hypoxia.

Parameters	Normoxia	Acute Hypoxia	Chronic Hypoxia	Chronic Hypoxia to Normoxia
AUC_(0–24)_, h·ug/L	581.65 ± 160.97	649.88 ± 97.27	1187.88 ± 216.95 ^ab^	926.65 ± 157.68 ^abc^
AUC_(0–∞)_, h·ug/L	613.97 ± 161.62	692.88 ± 120.78	1308.92 ± 241.35 ^ab^	994.02 ± 153.67 ^abc^
AUMC_(0–24)_, h^2^·ug/L	4243.09 ± 1550.07	4196.97 ± 573.20	9506.62 ± 2176.34 ^ab^	7653.48 ± 1632.40 ^ab^
AUMC_(0–∞)_, h^2^·ug/L	6180.89 ± 2829.55	5881.39 ± 1509.27	14,668.03 ± 3880.91 ^ab^	11,619.75 ± 3562.03 ^ab^
MRT_(0–24)_, h	7.19 ± 0.68	6.52 ± 0.84	7.95 ± 0.72 ^ab^	8.22 ± 0.45 ^ab^
MRT_(0–∞)_, h	9.79 ± 1.73	8.48 ± 1.39	11.08 ± 1.36 ^b^	11.51 ± 1.81 ^ab^
t_1/2z_, h	5.49 ± 1.77	5.64 ± 2.05	6.68 ± 1.83	6.19 ± 1.03
T_max_, h	3.25 ± 0.46	2.88± 0.35	3.12 ± 0.64	3.62 ± 0.52 ^b^
CL_z_/F, L/h/kg	34.31 ± 7.70	29.68 ± 5.45	15.74 ± 2.89 ^ab^	20.53 ± 3.02 ^ab^
V_z_/F, L/kg	270.40 ± 111.22	232.04 ± 63.22	150.37 ± 42.16 ^a^	184.23 ± 45.28
λ_z_, 1/h	0.14 ± 0.05	0.14 ± 0.06	0.11 ± 0.04	0.12 ± 0.02
C_max_, ug/L	82.49 ± 27.35	98.11 ± 23.74	135.30 ± 28.58 ^ab^	107.07 ± 19.24 ^c^

Note: Data are expressed as the mean ± SD values (*n* = 8). ^a^
*p* < 0.05 compared to the normoxia group; ^b^
*p* < 0.05 compared to the acute hypoxia group; ^c^
*p* < 0.05 compared to the chronic hypoxia group.

**Table 8 pharmaceuticals-15-01303-t008:** Pharmacokinetic parameters of hydroxybosentan in rat plasma after exposure to high-altitude hypoxia.

Parameters	Normoxia	Acute Hypoxia	Chronic Hypoxia	Chronic Hypoxia to Normoxia
AUC_(0–24)_, h·ug/L	16.90 ± 3.33	26.10 ± 6.43 ^a^	48.12 ± 7.13 ^ab^	32.67 ± 10.41 ^ac^
AUC_(0–∞)_, h·ug/L	17.52 ± 3.51	27.37 ± 6.51 ^a^	51.27 ± 6.16 ^ab^	34.52 ± 11.72 ^ac^
AUMC_(0–24)_, h^2^·ug/L	109.00 ± 28.12	159.46 ± 48.60 ^a^	299.63 ± 37.11 ^ab^	183.96 ± 61.42 ^ac^
AUMC_(0–∞)_, h^2^·ug/L	151.43 ± 39.09	231.94 ± 98.27	424.66 ± 67.64 ^ab^	261.46 ± 135.34 ^ac^
MRT_(0–24)_, h	6.41 ± 0.56	6.16 ± 1.34	6.27 ± 0.46	5.65 ± 0.67
MRT_(0–∞)_, h	8.67 ± 1.56	8.62 ± 3.26	8.41 ± 1.93	7.35 ± 1.94
t_1/2z_, h	5.04 ± 2.66	5.60 ± 2.30	5.55 ± 2.33	5.28 ± 3.23
T_max_, h	3.25 ± 0.46	3.00 ± 0.93	2.62 ± 0.52	3.25 ± 0.46
CL_z_/F, L/h/kg	1183.16 ± 239.74	766.53 ± 176.85 ^a^	395.84 ± 54.95 ^ab^	667.57 ± 322.49 ^ac^
V_z_/F, L/kg	8636.86 ± 5128.39	6360.02 ± 3074.79	3221.34 ± 1700.78 ^a^	4577.55 ± 2353.37
λ_z_, 1/h	0.17 ± 0.08	0.15 ± 0.09	0.15 ± 0.07	0.18 ± 0.11
C_max_, ug/L	3.57 ± 1.06	5.22 ± 1.43	7.92 ± 1.49 ^ab^	6.79 ± 2.51 ^a^

Note: Data are expressed as the mean ± SD values (*n* = 8). ^a^
*p* < 0.05 compared to the normoxia group; ^b^
*p* < 0.05 compared to the acute hypoxia group; ^c^
*p* < 0.05 compared to the chronic hypoxia group.

**Table 9 pharmaceuticals-15-01303-t009:** Linear gradient elution procedure of simvastatin and simvastatin hydroxy acid.

Time (min)	Acetonitrile	0.2% Formic Acid-Water
0.0	5	95
5.0	5	95
10.0	100	0
10.1	5	95
12.0	5	95

**Table 10 pharmaceuticals-15-01303-t010:** Linear gradient elution procedure of sildenafil and N-desmethyl sildenafil.

Time (min)	Acetonitrile	0.2% Formic Acid-Water
0.0	5	95
5.0	5	95
10.0	100	0
10.1	5	95
12.0	5	95

**Table 11 pharmaceuticals-15-01303-t011:** Linear gradient elution procedure of nifedipine and dehydronifedipine.

Time (min)	Acetonitrile	0.2% Formic Acid-Water
0.0	5	95
3.0	5	95
5.0	95	5
7.0	95	5
7.1	5	95

**Table 12 pharmaceuticals-15-01303-t012:** Linear gradient elution procedure of bosentan and hydroxybosentan.

Time (min)	Acetonitrile	0.2% Formic Acid-Water
0.0	10	90
1.0	10	90
6.0	100	0
7.0	100	0
7.1	10	90

## Data Availability

Data is contained within the article and Appendix A.

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
