# Peer review of "High-altitude Hypoxia Influences the Activities of the Drug-Metabolizing Enzyme CYP3A1 and the Pharmacokinetics of Four Cardiovascular System Drugs"

_pharmaceuticals, 2022, doi:10.3390/ph15101303_

Round 1

Reviewer 1 Report

Manuscript “High-altitude Hypoxia Influences the Activities of the Drug-Metabolizing Enzyme CYP3A1 and the Pharmacokinetics of Four Cardiovascular System Drugs” is a study in mice on the influence of hypoxia on the metabolism of cardiovascular drugs: Simvastatine, Bosentan, Nifedipine, and sildenafil.

This study is an important contribution; however, a few questions need addressing before publishing.

Major

There is no mention of the statistical analysis details, i.e., how the significance has been calculated remains unclear. In this regard, the significance levels indicated in Figures 1 and 2 are doubtful; in some cases, there is no difference assuming symmetrical error distribution. For example, white blood cell count in acute hypoxia compared to normoxia group (Figure 1).

For example, in Table 5, the significance of 99% was reported for several PK parameters with an overlapping standard deviation. Perhaps authors want to use a 90% confidence interval instead of 99%? A similar question is related to the results in Figure 13.

The study was not carried out in humans; therefore, it is impossible to extend these findings unconditionally to human patients (lines 545 and further).

Minor

Table 2. Errors are missing with some values

Author Response

Response to Reviewer 1 Comments

Point 1: There is no mention of the statistical analysis details, i.e., how the significance has been calculated remains unclear. In this regard, the significance levels indicated in Figures 1 and 2 are doubtful; in some cases, there is no difference assuming symmetrical error distribution. For example, white blood cell count in acute hypoxia compared to normoxia group (Figure 1). For example, in Table 5, the significance of 99% was reported for several PK parameters with an overlapping standard deviation. Perhaps authors want to use a 90% confidence interval instead of 99%? A similar question is related to the results in Figure 13. 

Response 1: Thanks very much for taking your time to review this manuscript. We really appreciate all your comments and suggestions. As you said, the significance levels of some data is not clearly described. We have unified the significance level values and P < 0.05 was considered to indicate a statistically significant difference in this article. All numerical data were expressed as mean ± SD. Oneway analysis of variance (ANOVA) was used to analyze Physiologic and Biochemical Parameters in rats. Comparisons between two groups were performed using the least significant difference(LSD). The Dunnett's t test was used for variance nonhomogeneity. All the statistical analyses were performed with the Statistical Package for the Social Sciences, version 20.0 (SPSS, Chicago, IL). The Homogeneity of variance test of Physiologic and Biochemical Parameters are sent in an Webpage attachment(Only the variance of the total protein was non-homogeneity, so Dunnett's t test was used).

P < 0.01” has been revised into “P < 0.05” in the Table 1-8, Figure 12-13 and in the results part 2.4 Pharmacokinetics of the manuscript. The sentence “Data are expressed as mean ± standard deviation (SD). Statistical differences were determined using one-way analysis of variance (ANOVA) on SPSS 20.0 software. Comparisons between two groups were performed using the least significant difference, and means were considered statistically significant at p < 0.05.” has been modified into “All numerical data were expressed as mean ± standard deviation (SD). Statistical differences were determined using one-way analysis of variance (ANOVA). Comparisons between two groups were performed using the least significant difference(LSD). The difference of P < 0.05 was considered statistically significant. The Dunnett's t test was used for variance nonhomogeneity. All the statistical analyses were performed with the Statistical Package for the Social Sciences, version 20.0 (SPSS, Chicago, IL).” in the Materials and Methods part 4.13 Statistical Analysis of the manuscript.

Point 2: The study was not carried out in humans; therefore, it is impossible to extend these findings unconditionally to human patients (lines 545 and further).

Response 2: Admittedly, our study was not carried out in humans, and it is impossible to extend these findings unconditionally to human patients. Therefore, the pharmacokinetics of nifedipine, bosentan, simvastatin, sildenafil, and their respective main metabolites in humans should be determined in our future study, which will make the study more complete and more rigorous. Thanks to your great idea.

The sentence “high-altitude hypoxia suppressed the metabolism of the drugs, indicating the need to adjust the dosage of the drugs accordingly in patients living in high-altitude areas to prevent drug accumulation and toxicity.” has been modified into “high-altitude hypoxia suppressed the metabolism of the drugs, indicating the pharmacokinetics of the drugs should be reexamined, and the optimal dose should be reassessed in patients living in high-altitude areas.” in the Abstract of the manuscript. The sentence “Therefore, we proposed that the dosages of nifedipine, bosentan, simvastatin, and sildenafil should be adjusted (reducing the dosage or increasing the dosing interval) in hypoxic environments for effective results.” has been been deleted in the fourth paragraph of the Discussion. The sentence “In summary, the findings of the present study showed that the pharmacokinetics of nifedipine, bosentan, simvastatin, and sildenafil in rats, including MRT, t1/2z, CLz/F, and Vz/F, were significantly affected by exposure to simulated high-altitude hypoxia, indicating the need for drug dosage adjustment in high-altitude hypoxic environments. Additionally, it is recommended that nifedipine, bosentan, simvastatin, or sildenafil should be administered to patients living in high altitudes environments in small doses and at close intervals to facilitate drug clearance from the system and avoid toxicity. However, further pharmacokinetic studies are necessary to determine the optimal dosage of nifedipine, bosentan, simvastatin, and sildenafil in high-altitude environments.” has been modified into “In summary, the findings of the present study showed that the pharmacokinetics of nifedipine, bosentan, simvastatin, and sildenafil in rats, including MRT, t1/2z, CLz/F, and Vz/F, were significantly affected by exposure to high-altitude hypoxia. Pharmacokinetic studies in humans are necessary to determine the optimal dosage of nifedipine, bosentan, simvastatin, and sildenafil in high-altitude environments.” in the last paragraph of the Discussion.

Point 3: Table 2. Errors are missing with some values.

Response 3: We are very sorry for the format error. Some data in Table 1 and Table 7 is hidden because the widths of the columns are too narrow for the content. We have adjusted the column width of Table 1 and Table 7 to ensure that all data can be displayed.

Reviewer 2 Report

Review to „High-altitude Hypoxia Influences the Activities of the Drug-Metabolizing Enzyme CYP3A1 and the Pharmacokinetics of Four Cardiovascular System Drugs”

The authors have presented an intensive study, comparing the pharmacokinetics of several drugs in normoxic conditions and high-altitude based hypoxia.  The manuscript is well written, still the results are presented in great detail, which make the overall understanding quite difficult. The authors should consider focusing more on the most important aspects and presenting side-plots in the supplemental material, if possible.

Besides few minor issues which could improve the overall clarity of the manuscript, there are three issues in the material and methods section that have to be clarified in order to put the presented data in the correct framework.

Having clarified these issues, the manuscript is in a good state and I can recommend the paper for publication after corrections.

Major Issue:

In the materials and methods section, the authors describe the individual experiments, however, the fate of the corresponding animal groups cannot be followed. From my understanding, there are two sites for this experiment, one in Xi’An (normoxic), one in Qinghai (high-altitude). Apparently, these two places are about 1000km distance apart. Therefor, there is considerable travel time involved for these animals. This is however not clear to the reader. Traveling long distances can have major impacts on the overall stress-level and as such also on the general metabolism of the animals and has be taken into account and must be reported. It is not clear, where the animals were sacrificed and the tissue was collected and processed. So the author have to amend the material and methods section accordingly: How did the animals reach the other experimental site, what was the travel route (car, plane,..) and duration. Where were the animals killed? At high-altitude or have they been transported back into the normoxic-laboratory? please also discuss the potential influences of the transport. If the samples were collected at different laboratories (normoxic vs high-altittude) – how did the authors ensure correct sample storage (temperature) until the next processing steps? I encourage the authors to critically revise this section with particular focus on the animal/sample transport between the two sites of the experiments as in its current presentation their results cannot be put into the correct framework.

Also, materials and methods: Please also clarify – what was the tissue preparation time from killing the animals until freezing / fixation of the tissue? Esp in hypoxia studies, this is critical as extended preparation times may lead to altered data.  (lines 599 ff)

Lines 607ff: Please clarify how the blood samples were collected. Did the authors anesthetize the rats, were they put in restrainiers? Are there any confounding factors in regard to the analysis of blood parameters, enzyme activities caused by stress or anesthesia?

Minor issues:

First paragraph: The reader would benefit from a clear definition of “high-altitude”, i.e. how many meters above sea level do the authors consider “high-altitude”. It would also put the later mentioned hight of 3 km into perspective.

2nd Paragraph, lines 61ff: Based on few examples, the authors state that “most drugs” are significantly affected. Please be more precise here, as this generalization based on 4 compounds seems injustified. Specifically, as the authors mention later, only a few enzymes are responsible for the metabolism of more than 90% of the drugs. Changing the order would straighten the story line in a way that this generalization after the introduction of the CYP450 enzymes, which are apparently affected by hypoxia, seems justified. Also, a clear statement on the motivation, why specifically these cardiovascular drugs are of extensive importance would strengthen the introduction.

In this reviewer’s opinion, the tables are overcrowded with unnecessary (in)significant digits. Often, the second or third significant digits are a pure result of mathematical calculations and are not always backed by the accuracy of the measurements themselves. I suggest to revise the tables and reduce the presented decimal placed to 1 or 0 where possible. (e.g. having an AUC of ~289 with a SD of 77 – do the measurements really provide accuracy to the mili h*ug/L ? do the decimal places really carry important information in these cases?),

Author Response

Response to Reviewer 2 Comments

Point 1: In the materials and methods section, the authors describe the individual experiments, however, the fate of the corresponding animal groups cannot be followed. From my understanding, there are two sites for this experiment, one in Xi’An (normoxic), one in Qinghai (high-altitude). Apparently, these two places are about 1000km distance apart. Therefor, there is considerable travel time involved for these animals. This is however not clear to the reader. Traveling long distances can have major impacts on the overall stress-level and as such also on the general metabolism of the animals and has be taken into account and must be reported. It is not clear, where the animals were sacrificed and the tissue was collected and processed. So the author have to amend the material and methods section accordingly: How did the animals reach the other experimental site, what was the travel route (car, plane,..) and duration. Where were the animals killed? At high-altitude or have they been transported back into the normoxic-laboratory? please also discuss the potential influences of the transport. If the samples were collected at different laboratories (normoxic vs high-altittude) – how did the authors ensure correct sample storage (temperature) until the next processing steps? I encourage the authors to critically revise this section with particular focus on the animal/sample transport between the two sites of the experiments as in its current presentation their results cannot be put into the correct framework.

Response 1: We appreciate your detailed suggestion of the manuscript. As you said, the way rats travel, where the rat tissue was sampled and storage methods of samples are not clearly described in this manuscript. The rats in high-altitude groups were airlifted from Xi'an to Xining City by plane and then transported by bus to Maduo County, the whole trip was completed in 10 hours. The rats in chronic hypoxia to normoxia group were transported by bus from Maduo to Xi'an, the whole trip was completed in 24 hours. Sample collection of the normoxia and chronic hypoxia to normoxia groups were carried out at Xi'an Jiaotong University Medical College, whereas the sample collection of the acute hypoxia and chronic hypoxia groups was performed at Maduo County Hospital. Hematological and Serum Biochemical Parameters were performed immediately after sampling and the tissues for Hematoxylin-eosin staining were fixed immediately. To determine the the protein and mRNA expression of CYP3A1 in rat liver, all liver samples were stored in liquid nitrogen and were transported to the Research Center for High Altitude Medicine of Qinghai University by bus. To determine the pharmacokinetics of drugs, all plasma samples were stored in liquid nitrogen and were transported to the State Key Laboratory of Plateau Ecology and Agriculture of of Qinghai University by bus. We have added the above content in the Materials and Methods of the manuscript. The effects of long-distance travel on the metabolism of drugs and CYP450 in rats needs further study to confirm. Thanks to your great idea, we will continue to conduct more research on this in the future.

The sentence “Rats in the acute hypoxia and chronic hypoxia groups received a 72-h acute hypoxia exposure and a 30-day chronic hypoxia exposure, respectively, at an altitude of 4,300 m and oxygen partial pressure of 11.1 kPa in Maduo, Qinghai Province. Rats in the chronic hypoxia to normoxia group were returned to Xi'an and raised for 7 days after a 30-day chronic hypoxia exposure in Maduo.” has been modified into “Rats in the acute hypoxia and chronic hypoxia groups received a 72-h acute hypoxia exposure and a 30-day chronic hypoxia exposure, respectively, at an altitude of 4,300 m and oxygen partial pressure of 11.1 kPa in Maduo, Qinghai Province. The rats in these two groups were airlifted from Xi'an to Xining City by plane and then transported by bus to Maduo County, the whole trip was completed in 10 hours. Rats in the chronic hypoxia to normoxia group were returned to Xi'an and raised for 7 days after a 30-day chronic hypoxia exposure in Maduo. The rats in chronic hypoxia to normoxia group were transported by bus from Maduo to Xi'an, the whole trip was completed in 24 hours. Sample collection of the normoxia and chronic hypoxia to normoxia groups were carried out at Xi'an Jiaotong University Medical College, whereas the sample collection of the acute hypoxia and chronic hypoxia groups was performed at Maduo County Hospital. Hematological and Serum Biochemical Parameters were performed immediately after sampling and the tissues for Hematoxylin-eosin staining were fixed immediately. The time from tissue sampling to putting the tissue into paraformaldehyde solution completed within 1 minute. To determine the protein and mRNA expression of CYP3A1 in rat liver, all liver samples were stored in liquid nitrogen and the time from liver sampling to putting the liver into liquid nitrogen completed within 1 minute. The liver samples were transported to the Research Center for High Altitude Medicine of Qinghai University by bus. To determine the pharmacokinetics of drugs, all plasma samples were stored in liquid nitrogen and were transported to the State Key Laboratory of Plateau Ecology and Agriculture of Qinghai University by bus.” in the Materials and Methods part 4.2 Animals and Experimental Treatments of the manuscript.

Point 2: Also, materials and methods: Please also clarify – what was the tissue preparation time from killing the animals until freezing / fixation of the tissue? Esp in hypoxia studies, this is critical as extended preparation times may lead to altered data.  (lines 599 ff)

Response 2: Hematological and Serum Biochemical Parameters were performed immediately after sampling and the tissues for Hematoxylin-eosin staining were fixed immediately. The time from tissue sampling to putting the tissue into paraformaldehyde solution completed within 1 minute. To determine the protein and mRNA expression of CYP3A1 in rat liver, all liver samples were stored in liquid nitrogen and the time from liver sampling to putting the liver into liquid nitrogen completed within 1 minute. The liver samples were transported to the Research Center for High Altitude Medicine of Qinghai University by bus. We have added the above content in the Materials and Methods of the manuscript.

Point 3: Lines 607ff: Please clarify how the blood samples were collected. Did the authors anesthetize the rats, were they put in restrainiers? Are there any confounding factors in regard to the analysis of blood parameters, enzyme activities caused by stress or anesthesia?

Response 3: Rats in each group were anesthetized via intraperitoneal injection of 20% urethane before the blood sampling. 0.3ml whole blood was collected from the ophthalmic venous plexus in anticoagulant tubes containing EDTA-2Na to determine Blood Physical Parameters. 0.5ml whole blood was collected from the ophthalmic venous plexus in the tube without anticoagulant, and then centrifuged and extracted the serum to determine Blood Biochemical Parameters. We have added the above content in the Materials and Methods of the manuscript.

Urethane-chloralose has shown to induce a decrease in basal blood pressure without exerting significant changes in heart rate (Le Noble et al., 1987). Urethane-chloralose also activates the sympathetic nervous system (Carruba et al., 1987) and might change blood perfusion to different tissues, including the liver (Gumbleton et al., 1990). Moreover, different authors have described that the use of urethane at high doses could affect drug metabolism through an inhibition of cytochrome CYP3A (Loch et al., 1995; Meneguz et al., 1999). The population PK modeling after FCZ and VRC intravenous dosing to Wistar rats showed that urethane anesthesia did not modify the PK behavior of both antifungals, two drugs with different routes of elimination, indicating that this anesthetic can be useful in preclinical PK investigations when animals have to be anesthetized(Francine Johansson Azeredo et al., 2015). Urethane anaesthesia did not modify pharmacokinetic behaviour of carvedilol in both normotensive and NG-nitro-L-arginine methyl hypertensive rats (Facundo Martín Bertera et al., 2015). Therefore, the effect of urethane on drug metabolism is still unclear. In addition, in the present study, the different groups of rats were anaesthetised in parallel, and to some extent, the effect of these factors on the results can be ignored. Based on your review, we reviewed the relevant literatures and found that there were very few studies focusing on the effect of urethane anaesthesia on blood parameters in rats. The effects of anesthesia on blood parameters, CYP50 activities and drug metabolism will be taken into account in our future study, which will make the study more complete and more rigorous. Thanks to your great idea.

References:

Le Noble, J. L., Struyker-Boudier, H. A., & Smits, J. F. (1987). Differential effects of general anesthetics on regional vasoconstrictor responses in the rat. Archives internationales de pharmacodynamie et de thérapie, 289, 82−92.

Carruba, M. O., Bondiolotti, G., Picotti, G. B., Catteruccia, N., & Da Prada, M. (1987). Effects of diethyl ether, halothane, ketamine and urethane on sympathetic activity in the rat. European Journal of Pharmacology, 134, 15−24. doi: 10.1016/0014-2999(87)90126-9

Gumbleton, M., Nicholls, P. J., & Taylor, G. (1990). Differential influence of laboratory anaesthetic regimens upon renal and hepatosplanchnic haemodynamics in the rat. Journal of Pharmacy and Pharmacology, 42, 693−697. doi: 10.1111/j.2042-7158.1990.tb06561.x

Loch, J. M., Potter, J., & Bachmann, K. A. (1995). The influence of anesthetic agents on rat hepatic cytochromes P450 in vivo. Pharmacology, 50, 146−153. doi: 10.1159/000139276. 

Meneguz, A., Fortuna, S., Lorenzini, P., & Volpe, M. T. (1999). Influence of urethane and ketamine on rat hepatic cytochrome P450 in vivo. Experimental and Toxicological Pathology, 51, 392−396. doi: 10.1016/S0940-2993(99)80027-X.

Francine Johansson Azeredo, Sandra Elisa Hass, Pedro Sansone, Hartmut Derendorf, Teresa Dalla Costa, Bibiana Verlindo De Araujo. (2015). Does the Anesthetic Urethane Influence the Pharmacokinetics of Antifungal Drugs? A Population Pharmacokinetic Investigation in Rats. J Pharm Sci. 104(10):3314-8. doi: 10.1002/jps.24552

Facundo Martín Bertera, Carla Andrea Di Verniero, Marcos Alejandro Mayer, Guillermo Federico Bramuglia, Carlos Alberto Taira, Christian Höcht. (2009). Is urethane-chloralose anaesthesia appropriate for pharmacokinetic-pharmacodynamic assessment? Studies with carvedilol. J Pharmacol Toxicol Methods. 59(1):13-20. doi: 10.1016/j.vascn.2008.10.001

The sentence “Routine blood examinations, including white blood cells, hemoglobin, red blood cells, hematocrit, mean corpuscular volume, platelet count, and mean platelet volume, were performed to determine the response of the rats to high altitudes conditions using the XN-10 automatic hematology analyzer (Sysmex Corporation, Japan).” has been modified into “Rats in each group were anesthetized via intraperitoneal injection of 20% urethane before the blood sampling. 0.3ml whole blood was collected from the ophthalmic venous plexus in anticoagulant tubes containing EDTA-2Na. Routine blood examinations, including white blood cells, hemoglobin, red blood cells, hematocrit, mean corpuscular volume, platelet count, and mean platelet volume, were performed to determine the response of the rats to high altitudes conditions using the XN-10 automatic hematology analyzer (Sysmex Corporation, Japan).” in the Materials and Methods part 4.3 Determination of Physiologic and Biochemical Parameters of the manuscript.

The sentence “The blood biochemical parameters of rats exposed to high-altitude hypoxia were examined, including alanine aminotransferase (ALT), aspartate transaminase (AST), total protein, albumin, globulin, urea, and glucose, using the AU5800 automatic biochemistry analyzer (Olympus Corporation, Japan).” has been modified into “ 0.5ml whole blood was collected from the ophthalmic venous plexus in the tube without anticoagulant, and then centrifuged and extracted the serum. The blood biochemical parameters of rats exposed to high-altitude hypoxia were examined, including alanine aminotransferase (ALT), aspartate transaminase (AST), total protein, albumin, globulin, urea, and glucose, using the AU5800 automatic biochemistry analyzer (Olympus Corporation, Japan).” in the Materials and Methods part 4.3 Determination of Physiologic and Biochemical Parameters of the manuscript.

Point 4: First paragraph: The reader would benefit from a clear definition of “high-altitude”, i.e. how many meters above sea level do the authors consider “high-altitude”. It would also put the later mentioned hight of 3 km into perspective.

Response 4: In 2004, the 6th International High Altitude Medical Conference was held in Xining, Qinghai Province, China. After full discussion, experts and scholars from all countries have determined that the plateau is elevations above 2500 meters above sea level (Rili Ge et al., 2015). High altitude hypoxia is believed to be experienced at elevations of more than 2500 meters above sea level (Nipa Basak et al., 2021). High altitude hypoxic stress is commonly faced by residents in areas with an average elevation exceeding 2500 meters and those who have just entered the plateau (Zi-ang Zhang et al., 2022). The sentence “High-altitude refers to the area where the altitude exceeds 2500 meters above sea level.” has been added in the section of Introduction.

References:

Rili Ge, Luobu Ouzhu, Junze Liu, et al. High Altitude Medical[M]. Beijing: Peking University Press, 2015.  

Nipa Basak, Tsering Norboo, Mohammed S Mustak, Kumarasamy Thangaraj. (2021). Heterogeneity in Hematological Parameters of High and Low Altitude Tibetan Populations. J Blood Med. 12: 287–298. doi: 10.2147/JBM.S294564

Zi-ang Zhang, Yafei Sun, Ziyan Yuan, Lei Wang, Qian Dong, Yang Zhou, Gang Zheng, Michael Aschner, Yuankang Zou, Wenjing Luo. (2022). Insight into the Effects of High-Altitude Hypoxic Exposure on Learning and Memory. Oxid Med Cell Longev. 2022: 4163188. doi: 10.1155/2022/4163188

Point 5: 2nd Paragraph, lines 61ff: Based on few examples, the authors state that “most drugs” are significantly affected. Please be more precise here, as this generalization based on 4 compounds seems injustified. Specifically, as the authors mention later, only a few enzymes are responsible for the metabolism of more than 90% of the drugs. Changing the order would straighten the story line in a way that this generalization after the introduction of the CYP450 enzymes, which are apparently affected by hypoxia, seems justified. Also, a clear statement on the motivation, why specifically these cardiovascular drugs are of extensive importance would strengthen the introduction.

Response 5: As you said, it is inaccurate to infer most drugs based on propranolol, aminophylline, acetazolamide, and sulfamethoxazole. The sentence “confirming that the metabolism of most drugs is slower in hypoxic environments” has been deleted in the second paragraph of the Introduction. According to your suggestion, we have adjusted the order of the second paragraph of the Introduction. First, it introduces CYP450, and points out that CYP3A4 metabolize more than 50% of prescription drugs, including simvastatin, sildenafil, bosentan, nifedipine. Then, it points out that hypoxia changes the expression of CYP3A4. Therefore, it is inferred that CYP3A4 activity may change under hypoxia and then affects the metabolism of these Four Cardiovascular System Drugs.

The sentence “Drug metabolic enzymes play key roles in the biotransformation of drugs, and their activities are considerably affected under hypoxic conditions[21-23]. Cytochrome P450 (CYP450) is the most important phase I metabolic enzyme, and 57 human CYP450 genes have been identified, which are classified into 18 families and 42 subfamilies according to sequence similarity[24-26]. CYP1A2, CYP2C9, CYP2C19, CYP2D6, CYP2E1, and CYP3A4 metabolize more than 90% of drugs, of which CYP3A4 is the most important drug metabolizing enzyme in the human body, metabolizing more than 50% of drugs[27-29]. Studies have demonstrated that hypoxia significantly alters the activity and expression of the drug-metabolizing enzyme CYP450[30-33], thereby affecting the biotransformation and pharmacokinetics of cardiovascular drugs such as simvastatin and nifedipine.” has been modified into “Cytochrome P450 (CYP450) is the most important phase I metabolic enzyme, and 57 human CYP450 genes have been identified, which are classified into 18 families and 42 subfamilies according to sequence similarity[21-23]. CYP1A2, CYP2C9, CYP2C19, CYP2D6, CYP2E1, and CYP3A4 metabolize more than 90% of drugs, of which CYP3A4 is the most important drug metabolizing enzyme in the human body, metabolizing more than 50% of prescription drugs[24-26]. simvastatin, sildenafil, bosentan, and nifedipine are metabolized by CYP3A4 in humans and by CYP3A1 in rats[27-31].Studies have demonstrated that hypoxia significantly alters the activity and expression of the drug-metabolizing enzyme CYP450[32-38]. Therefore, it is inferred that CYP3A4 activity may change under hypoxia and then affects the metabolism of these Four Cardiovascular System Drugs” in the second paragraph of the Introduction. We also changed the order of references.

Point 6: In this reviewer’s opinion, the tables are overcrowded with unnecessary (in)significant digits. Often, the second or third significant digits are a pure result of mathematical calculations and are not always backed by the accuracy of the measurements themselves. I suggest to revise the tables and reduce the presented decimal placed to 1 or 0 where possible. (e.g. having an AUC of ~289 with a SD of 77 – do the measurements really provide accuracy to the mili h*ug/L ? do the decimal places really carry important information in these cases?),

Response 6: Thanks very much for your suggestions. We have modified the significant digits of the data in the tables and reserved two digits after the decimal point.

Round 2

Reviewer 2 Report

The authors have provided missing information and clarified all issues. I believe the manuscript is acceptable for publication.